# Uncertainty Quantification in Retrieval Augmented Question Answering

**Laura Perez-Beltrachini**                                          *lperez@ed.ac.uk*
*University of Edinburgh*

**Mirella Lapata**                                                  *mlap@ed.ac.uk*
*University of Edinburgh*

**Reviewed on OpenReview:** https://openreview.net/forum?id=JLkgI0h7wy

## Abstract

Retrieval augmented Question Answering (QA) helps QA models overcome knowledge gaps by incorporating retrieved evidence, typically a set of passages, alongside the question at test time. Previous studies show that this approach improves QA performance and reduces hallucinations, without, however, assessing whether the retrieved passages are indeed useful at answering correctly. In this work, we propose to quantify the uncertainty of a QA model via estimating the utility of the passages it is provided with. We train a lightweight neural model to predict passage utility for a target QA model and show that while simple information theoretic metrics can predict answer correctness up to a certain extent, our approach efficiently approximates or outperforms more expensive sampling-based methods.[1]

## 1 Introduction

Retrieval augmented Question Answering (QA) allows QA models to overcome knowledge gaps at test time through access to evidence in the form of retrieved passages (Lewis et al., 2020; Guu et al., 2020; Izacard et al., 2024). Recent work leverages external retrievers (Chen et al., 2017; Izacard & Grave, 2021) and the language understanding and generation capabilities of Large Language Models (LLMs; Brown et al. 2020; Ouyang et al. 2024) to predict answers based on questions *and* retrieved passages which are provided as input context. In Figure 1, we show an example of a question (*Who sings Does He Love Me with Reba?*), retrieved passages, and predicted answers.

Retrieval augmented QA architectures have proven beneficial in increasing LLM performance on tail knowledge (Izacard et al., 2024; Mallen et al., 2023), reducing hallucinations in the generated answers (Shuster et al., 2021), and even improving model calibration (Jiang et al., 2021). However, there are various ways in which retrieval augmented QA can go wrong. The set of retrieved passages is far from perfect (Sciavolino et al., 2021; Yoran et al., 2024; Kasai et al., 2024) containing irrelevant, incomplete, or misleading evidence, the model might be under-trained to read certain passages and reason over these and the question (Izacard et al., 2024; Liu et al., 2024b; Sun et al., 2025), and the question can be ambiguous or unanswerable (Kasai et al., 2024). Ultimately, QA models may not follow the provided passages (Xie et al., 2024; Joren et al., 2025). When faced with uncertainty, QA models should ideally acknowledge it (e.g., by communicating it) rather than risk an incorrect response.

A good deal of previous work has focused on quantifying *answer uncertainty* in the context of *closed-book* QA tasks, where the answer is predicted based on a question and the model's encoded knowledge. Sampling-based methods rely on output discrepancies among multiple predictors of the same input (Gal & Ghahramani, 2016; Lakshminarayanan et al., 2017). They measure diversity on a set of answers (Kuhn et al., 2023; Chen & Mueller, 2024) sampled via temperature scaling (Guo et al., 2017), with larger variance indicating higher

---

[1]Code and data are available at https://github.com/lauhaide/ragu

Figure 1: Example question from Natural Questions dataset (Kwiatkowski et al., 2019) with two top-retrieved passages using Contriever-MSMARCO (Izacard et al., 2022). On top of each passage, we show the answer generated by Gemma2-9B when prompted with that passage and the question. The QA model answers correctly (green) only when prompted with the first passage. Numbers at the bottom right of each passage are utility scores predicted by our model (higher values indicate more useful passages).

uncertainty. LLM-based methods rely on the QA model's own judgment of uncertainty (Kadavath et al., 2022; Lin et al., 2022; Tian et al., 2023). Through prompting, the model is encouraged to express its uncertainty (e.g., 0.5 or '*almost certain*'), either alongside the predicted answer (Lin et al., 2022; Tian et al., 2023) or after generating it (Kadavath et al., 2022; Tian et al., 2023). None of these approaches has been applied in the context of *retrieval augmented* QA (Shorinwa et al., 2025).

In this paper, we focus on answer uncertainty estimation in the context of retrieval augmented QA. We hypothesize that a passage is *useful*, if a model can correctly answer questions based on it. If passages are informative and prime the QA model towards appropriate knowledge Geva et al. (2021), we expect it to produce a correct answer. On the contrary, if passages are irrelevant or misleading and the question falls outside the QA model's knowledge, it is likely to produce an incorrect answer, either factually inaccurate or entirely fabricated. Importantly, this notion of *utility* is based on how the target QA model will answer with the provided passages and not on what an external judge (e.g., entailment model) thinks about them. We quantify the utility of a retrieved passage with a small neural model trained on utility judgments obtained by observing the target QA model's answering behavior. We borrow ideas from direct uncertainty quantification approaches (Van Amersfoort et al., 2020; Lahlou et al., 2023) but do not decompose uncertainty or outline shifts in the input distribution. We make utility predictions for each retrieved passage which we then use to estimate the uncertainty of the QA model when prompted with a set thereof.

We evaluate our approach on short-form information-seeking QA tasks (Rodriguez & Boyd-Graber, 2021) (see Figure 1 for an example). Results on six datasets show that our uncertainty estimator is comparable or outperforms existing sampling-based methods while being more test-time efficient. Sampling-based solutions are expensive for in-production QA systems, in terms of latency and cost (e.g., QA engines built on top of proprietary language models would need to process relatively long prompts). Moreover, our experiments reveal that variation is less prominent in model answers in the context of retrieval augmented QA (e.g., the QA model is more confident on incorrect answers supported by retrieved passages in the prompt). Our contributions can be summarized as follows:

- We quantify QA model uncertainty via estimating the utility of the passages it is provided with.
- We (contrastively) train a small neural model on utility scores obtained through combining accuracy (is the generated answer correct?) and entailment (is the generated answer supported by the passage?) metrics.
- Our approach is lightweight, improves upon more expensive sampling-based methods, and is not tied to the retriever (and passages) used to prompt the QA model.

## 2 Related Work

**Uncertainty Quantification for Question Answering** Several methods have been proposed to predict answer uncertainty in the context of closed-book QA; however, none of them has analysed uncertainty in retrieval augmented QA models. Many of these methods rely on the assumption that variation in the model's output is an expression of its uncertainty (Kuhn et al., 2023; Farquhar et al., 2024; Chen & Mueller, 2024). For example, some approaches (Kuhn et al., 2023; Farquhar et al., 2024) first cluster answers with similar meaning (in a sample) via Natural Language Inference (NLI) before computing entropy while other work (Chen & Mueller, 2024) focuses on *black-box* models; they also compute similarities in the set of answers but associate them with the model's self-judgement of confidence. In a recent study, Soudani et al. (2025) show that, in the context of retrieval augmented QA, these methods tend to exhibit overconfidence and sensitivity to the input context. Indeed, our experiments corroborate their analysis. We find that sampling-based methods exhibit less variation in a retrieval setting, and do not offer a significant advantage over methods such as perplexity or LLM prompting for self-assessment (Kadavath et al., 2022), unlike in closed-book settings.

Sequence entropy methods (Kuhn et al., 2023; Farquhar et al., 2024) focus on detecting incorrect answers stemming from arbitrary fluctuations in model outputs (referred to as confabulations). Our approach extends to additional error sources, including incorrect training data or misleading evidence. Hou et al. (2024) focus on quantifying aleatoric uncertainty (i.e., uncertainty in the data) caused by ambiguous questions, an approach that could be combined with ours. Sampling-based methods are expensive to run at inference time for a production QA system; they require several inference steps in addition to performing similarity computations, which can become more complex with longer answers (Zhang et al., 2024b). In contrast, our approach is light-weight at inference time, but requires training data with observations pertaining to the accuracy of the target model. Our approach optionally uses an NLI model at training time, depending on the chosen objective. In Section 6, we further summarise key differences between our approach and existing uncertainty estimation methods in the context of retrieval augmented QA.

**Judging the Quality of Retrieved Passages** Previous work has analysed the quality of retrieved passages (Yu et al., 2023; Asai et al., 2024; Wang et al., 2024; Xu et al., 2024; Yoran et al., 2024) as they can be irrelevant or misleading. Asai et al. (2024) make use of an external critic model to create training data exemplifying cases where a question requires retrieval (or not) and, in the case that retrieval is needed, whether retrieved passages contain the information (or not) to formulate the answer. Note that the usefulness judgment is made by an external critic. Then, a QA model (i.e., the Self-RAG model) is trained on this data to learn to reflect by itself whether passages are supportive and relevant and to predict special tokens indicating this. While it is possible to derive uncertainty from those special tokens' probabilities, they only reflect Self-RAG's uncertainty state. Our proposal is more general and aims at predicting answer uncertainty in zero-shot QA models (e.g., instruction fine-tuned LLMs). Thus, using Self-RAG special tokens would be like using an off-the-shelf sophisticated classifier (i.e., a specialized textual entailment) to predict the uncertainty of a zero-shot QA model (Yoran et al., 2024). In contrast, we predict uncertainty by observing the errors of the target QA model (in a zero-shot setting). Other work creates auxiliary tasks around retrieved passages enforcing the QA model to reason on them; e.g., by taking notes about each passage (Yu et al., 2023) or generating passage summaries (Xu et al., 2024). These methods also use extreme-scale LLMs to generate training data for *fine-tuning* a retrieval augmented QA model. Park et al. (2024) select low and high quality in-context examples in order to instruct the LLM to reason on input passages. Concurrent work by Joren et al. (2025) defines the concept of context sufficiency, i.e., whether the content in the set of retrieved passages is sufficient to answer the question and uses an external judge to assess this. However, the external judgment might not agree with the answering behavior of the target QA model (i.e., the judge may indicate that the context is sufficient but the model may still incorrectly answer). All these approaches aim at improving QA performance while our primary goal is modeling QA uncertainty.

**Using a Separate Model to Predict Confidence** Our passage utility predictor is related to methods aiming to estimate error *directly* (Lahlou et al., 2023), e.g., by training a secondary model to estimate target model loss; instead, our predictor is trained with sequence-level metrics, i.e., accuracy and entailment, which measure error *indirectly*. Some work (Kamath et al., 2020; Zhang et al., 2021) predicts answer correctness

in the context of Reading Comprehension (the task of generating an answer based on a single supportive passage). However, as there is no retrieval involved, the input passage is by default useful, and the main goal is to detect answer uncertainty due to the QA model being under-trained. In our setting, the number and quality of passages varies leading to additional sources of uncertainty (e.g., misleading information). Some approaches train a specific model to predict answer confidence scores (Dong et al., 2018; Kamath et al., 2020; Zhang et al., 2021; Mielke et al., 2022) by incorporating various features from the input and model output. Our approach predicts answer uncertainty directly from individual passage utilities and its predictions could also be combined with other features (e.g., output sequence probability).

## 3 Modeling Answer Uncertainty

We formally define retrieval augmented QA as follows. Given question $x$ and a set of retrieved passages $R = \{p_1, p_2, \cdots, p_{|R|}\}$ obtained with retriever $\mathcal{R}$, an LLM-based QA model $\mathcal{M}$ is prompted to generate answer $y$ to question $x$ token-by-token according to its predictive distribution:

$$P(y|x, R; \mathcal{M}) = \prod_{t=1}^{|y|} P(y_t|y_{1..t-1}, x, R; \mathcal{M}). \tag{1}$$

We wish to estimate $\mathcal{M}$'s uncertainty (i.e., chance of error) of generating $y$ given $x$ and $R$.

When a retrieved passage is useful to answer a given question (such as the first passage in Figure 1 for the question *Who sings Does He Love Me with Reba?*), the QA model is likely to be confident when generating the answer (*Linda Davis*). When the passage is not useful (such as the second passage in Figure 1), the QA model is likely to be uncertain and provide an incorrect answer (*Patti LaBelle*). Our hypothesis is that the utility of each passage $p$ in $R$ is indicative of the QA model's uncertainty in generating $y$, when prompted with $R$. If there are passages in $R$ with high utility (e.g., in Figure 1, the first passage is useful to answer the question), it is likely that the QA model will be confident when generating answer $y$. In contrast, if all passages in $R$ have low utility, it is likely that the QA model will be uncertain when generating the answer.

The core of our approach is estimating the utility $v_{\mathcal{M}}$ of individual passages for a target QA model $\mathcal{M}$. Specifically, we develop an estimator $\{x, p\} \mapsto v_{\mathcal{M}}(\{x, p\})$ for each passage $p \in R$ (Section 3.1). We then leverage the predicted passage utility $v_{\mathcal{M}}$ to estimate $\mathcal{M}$'s uncertainty when generating answer $y$ to question $x$ based on evidence $R$, $\{x, R\} \mapsto \mathbf{u}_{\mathcal{M}}(\{x, R\})$ (Section 3.2).

### 3.1 Passage Utility Prediction

Intuitively, a retrieved passage $p$ is useful for a QA model $\mathcal{M}$ if $\mathcal{M}$ can correctly answer question $x$ when prompted with $p$. However, the model's dependence on $p$ may vary. In some cases, $\mathcal{M}$ may generate the correct answer even if $p$ does not explicitly contain it, instead it positively primes the model to draw upon its memorized knowledge. In Figure 1, the first passage has high utility because the QA model generates a correct answer when prompted with it, and explicitly states that "Linda Davis sings alongside Reba McEntire". In contrast, the second passage, while related to the question's topic, is not useful. The QA model struggles to answer correctly when prompted with it, suggesting uncertainty. Since this passage does not provide helpful information and leads to incorrect answers, its utility is low.

We estimate the utility of passage $p$ in answering question $x$ for QA model $\mathcal{M}$ by combining two key measures, accuracy and entailment:

$$v_{\mathcal{M}} = \frac{(a(y) + e(y))}{2} \tag{2}$$

Accuracy, denoted as $a(y)$, indicates whether the generated answer $y$ is correct, while entailment, denoted as $e(y)$, measures the degree to which $p$ supports $y$. Accuracy is determined by an evaluator $A$, and entailment is assessed using an NLI model $E$. The combined passage utility $v_{\mathcal{M}}$ ranges between 0 and 1, given that $a$ takes values in $\{0, 1\}$ and $e$ falls within the $[0, 1]$ interval. The intuition behind this composite term $v_{\mathcal{M}}$ is to encourage a ranking of passages that spans a broad spectrum: from highly useful ones (producing correct

answers with strong entailment), through more ambiguous cases (e.g., correct answers with weak entailment or incorrect answers with strong entailment), to clearly unhelpful passages (producing incorrect answers with weak entailment).[2]

We train a lightweight neural model on dataset $D_{\mathcal{M}} = \{(x, p, v_{\mathcal{M}})\}$ to predict passage utility scores, $\{x, p\} \mapsto v_{\mathcal{M}}(\{x, p\})$, We construct $D$ by pairing input questions $x$ and passages $p$ with utility scores $v_{\mathcal{M}}$ which we obtain after running $\mathcal{M}$ on examples $(x, p)$ and computing observed answer accuracy and entailment scores from the QA model $\mathcal{M}$. We retrieve $|R| > 1$ passages per question to ensure a diverse range of usefulness and create training instances $\{(x, p_i, v_i) \,|\, p_i \in R\}$ under model $\mathcal{M}$. We leverage this data to train the passage utility predictor with a contrastive learning scheme. Specifically, if two passages $p_i$ and $p_j$ belong to $R$ and $p_i$ is more useful than $p_j$ for answering question $x$, the predicted utility score $v_i$ should be higher than $v_j$ by margin $m$, ensuring that $p_i$ is ranked above $p_j$. We train the utility predictor with the following ranking objective:

$$\mathcal{L}_{rank} = \sum_{((x,p_i),(x,p_j)) \in R \times R, i \neq j} \max(0, -z(v_i - v_j) + m)), \tag{3}$$

where $z$ controls the gold order between $p_i$ and $p_j$ (e.g., if $z = 1$, $p_i$ has higher utility, and conversely $z = -1$ indicates the opposite ordering) and $m$ is a hyper-parameter.

We train the passage utility predictor with a Siamese neural network consisting of a BERT-based encoder (Devlin et al., 2019) followed by pooling and two MLP layers stacked on top of BERT outputs (Fang et al., 2024). The output layer computes the utility score as $v_i = W_o h^L + b_o$ where $h^L$ is the vector representation for $(x, p_i)$ from the last hidden layer (the $L$-th layer) of the network. At inference time, we compute a single utility score for each passage. We provide implementation and training details in Section 4.

To strengthen the role of accuracy prediction as a training signal and regularize the range of utility values learned by the ranking scheme, we combine the ranking objective in Equation (3) with the following Binary Cross Entropy (BCE; Sculley 2010) objective:

$$\mathcal{L}_{BCE} = \sum_{((x,p_i),(x,p_j)) \in R \times R, i \neq j} [a \times \log(p(x, p_i)) + (1 - a) \times \log(1 - p(x, p_i))]$$
$$+ [a \times \log(p(x, p_j)) + (1 - a) \times \log(1 - p(x, p_j))], \tag{4}$$

where $p(x, p_i) = \mathrm{sigmoid}(v_i)$ and $a$ is the target accuracy label $a(y)$ under model $\mathcal{M}$ taking values in the set $\{0, 1\}$. We train the utility predictor with the following combined objective where $\lambda$ is a hyper-parameter to adjust the $\mathcal{L}_{BCE}$ penalty:

$$\mathcal{L} = \mathcal{L}_{rank} + \lambda \mathcal{L}_{BCE}. \tag{5}$$

Both ranking and BCE objectives are compatible with gold annotations that could be provided in active learning or interactive settings (Simpson et al., 2020; Fang et al., 2024). For example, moderators of the QA system would provide judgments on the accuracy of the answers it predicts (e.g., *correct/incorrect*) and the extent to which these are supported by the retrieved passages (e.g., *not* supported to *fully* supported).

## 3.2   Answer Uncertainty Estimation

For our QA task, we want to define an estimator $\{x, R\} \mapsto \mathbf{u}_{\mathcal{M}}(\{x, R\})$ which quantifies the uncertainty of model $\mathcal{M}$ when generating answer $y$ for question $x$ based on a prompt with passages $R$. We propose estimating $\mathbf{u}_{\mathcal{M}}$ directly from the utility scores of individual passages in $R$. The key intuition is that the higher the utility of one (or more) passages, the less uncertain $\mathcal{M}$ will be when generating answer $y$. Conversely, if all passage utilities in $R$ are low, it is more likely that $\mathcal{M}$ will be uncertain about the answer. Specifically, we estimate answer uncertainty $\mathbf{u}_{\mathcal{M}}$ by taking the maximum utility score among the passages in $R$:

$$\mathbf{u}_{\mathcal{M}}(\{x, R\}) = \max(v_{\mathcal{M}}(\{x, p\}) \,|\, p \in R). \tag{6}$$

---

[2]Note that for the pairwise ranking loss, using the sum $v_{\mathcal{M}} = a(y) + e(y)$ would be equivalent. In Appendix A, we include ablation experiments with different implementations of the Passage Utility score $v_{\mathcal{M}}$.

Our approach to aggregating passage utilities is intuitive and simple. However, QA models might be sensitive to factors relating to how they are prompted, such as the number or order of passages (Liu et al., 2024b; Xie et al., 2024). In Appendix C, we examine such confounds more closely, comparing question answering accuracy when models are prompted with individual passages in $R$ (our aggregation approach) versus the entire set $R$. The study shows that there is little disagreement between the two methods and that it is possible to approximate answer uncertainty when prompting with $|R|$ passages while avoiding the combinatorial complexity of estimating uncertainty over all possible combinations of input passages.

## 4 Experimental Setup

### 4.1 QA Tasks and Models

We evaluate our approach to predicting answer uncertainty on short-form question answering tasks. Specifically, on the following six datasets: Natural Questions (Kwiatkowski et al., 2019), TriviaQA (Joshi et al., 2017), WebQuestions (Berant et al., 2013), SQuAD (Rajpurkar et al., 2016), and PopQA (Mallen et al., 2023). We also evaluate on RefuNQ (Liu et al., 2024a), a dataset with unanswerable questions about non-existing entities. In Appendix F.1, we describe each dataset, provide example questions, and make available details about the splits used in our experiments which follow Lee et al. (2019).

We consider backbone instruction fine-tuned LLMs from different families of similar size. These are Llama-3.1-8B-Instruct (AI@Meta, 2024), Mistral-7B-Instruct-v0.3 (Jiang et al., 2023), and Gemma2-9B-it (Riviere et al., 2024). We also assess answer uncertainty quantification performance on QA models of the same family but with different sizes. To this end, we additionally evaluate on Gemma2-27B-it. For all QA models, we use a simple prompt including the retrieved passages and the question in the context; the prompt is shown in Table 14 in the Appendix. The QA models' answer is the most likely answer generated with greedy sampling at temperature equal to 0. Following previous work on retrieval augmented QA, we use Contriever-MSMARCO (Izacard et al., 2022) as our external retriever (Asai et al., 2024) and the target QA models are prompted with $|R| = 5$ passages (Yu et al., 2023; Asai et al., 2024; Xu et al., 2024). In Appendix F.2, we provide more details about inference and passage retrieval.

### 4.2 Evaluation

**QA Accuracy** A precise metric for measuring accuracy is key when evaluating the quality of uncertainty estimation. Token overlap metrics are imprecise and can over- or under-estimate accuracy (e.g., 5 will not match *five*). Thus, we rely on a LLM-based accuracy evaluator to create training data for the Passage Utility predictor (i.e., $A$ in Section 3.1) and to evaluate retrieval augmented QA performance. We use Qwen2-72B-Instruct (Yang et al., 2024) and the prompt proposed by Sun et al. (2024) to obtain accuracy judgments. Details about the LLM evaluator can be found in Appendix F.2.

**Uncertainty Estimation** To assess the quality of answer uncertainty prediction, we follow Farquhar et al. (2024) and report the Area Under the Receiver Operator Curve (**AUROC**) on detecting incorrect answers (i.e., answers with high uncertainty). In Appendix G.3, we also report the area under the 'rejection accuracy' curve (**AURAC**) which captures the accuracy a model would have if it refused to answer questions with highest uncertainty. Rejection accuracy is essentially the model's accuracy on the remaining questions. In the main results section, we focus on selective answering performance when models answer 80% of the least uncertain questions versus when always answering. We provide implementation details in Appendix F.2.

### 4.3 Methods

**Passage Utility Predictor** We train a Passage Utility predictor per QA model and QA task. For each task, we curate dataset $D_{\mathcal{M}} = \{(x, p, v_{\mathcal{M}})\}$ to train and evaluate a Passage Utility predictor for QA model $\mathcal{M}$. We use the training (and development) questions available for each QA task, considering the top five retrieved passages for each question (i.e., $|R| = 5$). Note that $|R|$ is a hyper-parameter and other values would be also possible. Larger sizes of $|R|$ would yield more training data, since the Utility predictor takes individual passages (together with the question) as input. The target QA model $\mathcal{M}$ is first prompted with

passage $p \in R$ and question $x$ to generate answer $y$. Then, we annotate passages $p$ with a utility score computed with the accuracy evaluator $A$ and entailment judge $E$ on generated answer $y$ (Section 3.1). We use an ALBERT-xlarge model (Lan et al., 2020) optimized on MNLI (Williams et al., 2018) and the VitaminC dataset (Schuster et al., 2021). We provide more details about the curated datasets and training of the Passage Utility predictor training in Appendix F.2.

**Comparison Approaches and Baselines**   There exist several uncertainty estimation methods which we group in two categories based on whether they require access to logits or simply model outputs (see Fadeeva et al. 2023 for additional methods). We choose the highest scoring ones to compare with here and include additional results in Appendix G.3 for completeness.

*Information-Theoretic Measures.* We compare against uncertainty estimation methods that are based on the predictive probabilities of the target QA model. Let $y$ denote an answer generated with probability $p(y|x, R; \mathcal{M})$ which is computed as:

$$p(y|x, R; \mathcal{M}) = \prod_{t=1}^{|y|} p(y_t|y_{1..t-1}, x, R; \mathcal{M}) \tag{7}$$

The **Perplexity** (PPL) of model $\mathcal{M}$ boils down to calculating *token-level* entropy as it is based on the average negative log-likelihood of the generated tokens:

$$\text{PPL}(x, R, \mathcal{M}) = \exp\left\{-\frac{1}{|y|} \sum_{t=1}^{|y|} \log\, p(y_t|y_{1..t-1}, x, R; \mathcal{M})\right\}, \tag{8}$$

**Regular entropy**, on the other hand, is computed over *sequences*, quantifying the entropy of the answers. It is defined as $\mathbb{E}[-\log P(Y|x, R; \mathcal{M})]$ where the expected value, $\mathbb{E}$, is computed on sequences $y$ sampled from the conditional distribution $P(Y|x, R; \mathcal{M})$, where random variable $Y$ denotes the answer sequences, and $x$ and $R$ are the input question and retrieved passages, respectively. In practice, regular entropy is approximated via Monte-Carlo integration, i.e., sampling $N$ random answers from $P(Y|x, R; \mathcal{M})$:

$$\text{RE}(x, R, \mathcal{M}) = -\frac{1}{N} \sum_{n=1}^{N} \log \tilde{P}(y^n \mid x, R; \mathcal{M}), \tag{9}$$

where $\tilde{P}(y^n \mid x, R; \mathcal{M})$ is the length normalised version of $P(y^n|x, R; \mathcal{M})$.

Kuhn et al. (2023) propose **Semantic Entropy**, a variant of regular entropy that disregards uncertainty related to the surface form of the generated answers. The method works by sampling several possible answers to each question and grouping the set of $N$ samples into $M$ clusters (with $M \leq N$) that have similar meanings (which are determined on the basis of whether answers in the same cluster entail each other bidirectionally). The average answer probability within each cluster is:

$$\text{SE}(x, R, \mathcal{M}) = -\sum_{m=1}^{M} \hat{P}_m(x, \mathcal{M}) \log \hat{P}_m(x, \mathcal{M}), \tag{10}$$

where $\hat{P}_m(x, \mathcal{M})$ is estimated as follows:

$$\hat{P}_m(x, \mathcal{M}) = \frac{\sum_{y \in C_m} \tilde{P}(y|x, R; \mathcal{M})}{\sum_{m=1}^{M} \sum_{y \in C_m} \tilde{P}(y|x, R; \mathcal{M})} \tag{11}$$

*LLM-based Measures.* We compare with **p(true)** which uses the same LLM-based target QA model to assess whether the answers it produces are correct (Kadavath et al., 2022). We follow the p(true) variant used in previous work (Kuhn et al., 2023). The QA model is prompted with the question and a set of candidate answers (consisting of the most likely answer and a sample of $N$ answers) and is instructed to respond whether the most likely answer is true or false (i.e., correct/incorrect). The score produced by this approach is the

probability of model $\mathcal{M}$ generating the token True. This method needs several in-context examples to work well; following Kuhn et al. (2023), we use 20 in-context examples. Note that since our backbone LLMs are recent models with a large context (unlike Kuhn et al. 2023), all 20 examples are fed in the context making p(true) an expensive but very strong approach. In the context of retrieval augmented QA, we modify p(true) to include in the prompt the set of retrieved passages for the question of interest. We provide the prompt used by p(true) in Appendix F.3. Note that p(true) can be considered as a specialized powerful entailment model and thus we do not include entailment based methods relying on off-the-shelf NLI models which have been shown to perform poorly (Yoran et al., 2024).

For approaches that require sampling, we follow previous work (Farquhar et al., 2024) and take $N = 10$ samples, which we generate with multinomial sampling. We set the sampling temperature to 1, with nucleus sampling ($P = 0.9$; Holtzman et al. 2020) and top$-K$ sampling ($K = 50$; Fan et al. 2018), and use a different random seed to draw each sample. We provide further details about inference in Appendix F.2 and report inference costs for each approach in Appendix B.

## 5    Results and Analysis

**Passage Utility is effective across model families, sizes, and QA tasks.**    Table 1 summarizes our uncertainty estimation results (test set) with four QA models (Gemma2-9B/27B, Llama-3.1-8B, and Mistral-7B-v0.3) across six QA datasets (results on the development set are included in Appendix G.3). We boldface the highest AUROC value for each QA model and dataset pair and mark with * the next best value that is significantly different from it at $p < 0.05$. We use the paired De Long test (DeLong et al., 1988) to compute whether two AUROC values are significantly different.[3]

In general, answer perplexity (PPL) performs rather poorly, especially with Gemma2-9B/27B. Perplexity is likely to underperform with less calibrated models, such as those which have undergone instruction tuning (Tian et al., 2023). Regular Entropy shows little improvement upon PPL but by ignoring surface form choices and focusing on meaning, Semantic Entropy improves AUROC scores. p(true) performs well at detecting answer uncertainty matching or surpassing Semantic Entropy. Overall, we observe that the gap among these methods' performance is lower than in the context of closed-book QA studied in previous work (Farquhar et al., 2024). We hypothesize that, on one hand, our recent QA models admitting more in-context examples benefiting p(true) and, on the other hand, that retrieved passages in the prompt make QA models' outputs less varied. Our Passage Utility approach performs on par or outperforms all other methods with a *single small-model inference step* on each input passage.

Passage Utility performs particularly well on challenging question answering tasks represented by datasets like PopQA and RefuNQ. In these cases, our light-weight uncertainty estimation model works better than p(true) which requires the same QA model (i.e., the same backbone LLM) to judge the correctness of its own generated answers. We speculate that for questions with high uncertainty, i.e., where the model does not have the knowledge to answer (e.g., questions about non-existing concepts in RefuNQ), it confidently generates a response and also fails at assessing it. We attribute the Passage Utility's success to the fact that it has been specifically trained to detect situations where the target QA model is prone to answer incorrectly (i.e., when provided with retrieved passages of lower relevance). The six QA tasks pose different retrieval challenges. On TQA, retrieval results are often of good quality: for 73% of the questions, the top-5 retrieved passages contain the gold answer string. In contrast, on PopQA the percentage reduces to 63% and on RefuNQ the quality of retrieval is deliberately low (as it consists of unanswerable questions). Across models (Table 1), our approach is comparable to p(true) when retrieved passages contain the answer and excels in cases of low quality retrieval.

Passage Utility also performs well with different QA model sizes (within the same family), i.e., Gemma2-9B and 27B (Table 1). We observe a noticeable decrease in performance for most information-theoretic models when using the biggest Gemma model (27B). We attribute this to the fact that the 27B model more confidently makes less errors and its calibration may be affected more by the fine-tuning step (Tian et al., 2023). p(true), on the other hand, benefits from the largest model's context understanding and memorized knowledge.

---

[3]We use the library in `https://github.com/Brritany/MLstatkit` to compute significance scores.

| | Gemma2-9B | | | | | | | Gemma2-27B | | | | | | |
|---|---|---|---|---|---|---|---|---|---|---|---|---|---|---|
| | NQ | TQA | WebQ | SQuAD | PopQA | RefuNQ | AVG | NQ | TQA | WebQ | SQuAD | PopQA | RefuNQ | AVG |
| PPL | 0.64 | 0.68 | 0.52 | 0.53 | 0.59 | 0.51 | 0.58 | 0.64 | 0.50 | 0.53 | 0.59 | 0.58 | 0.51 | 0.56 |
| p(true) | 0.73 | 0.75* | 0.67 | 0.63 | 0.81* | 0.62 | 0.70 | **0.77** | **0.83** | 0.67 | 0.68* | 0.78* | 0.60* | 0.72 |
| Regular Entropy | 0.66 | 0.69 | 0.54 | 0.56 | 0.61 | 0.51 | 0.60 | 0.67 | 0.54 | 0.55 | 0.59 | 0.62 | 0.51 | 0.58 |
| Semantic Entropy | 0.70 | 0.73 | 0.57* | 0.64* | 0.73 | 0.59* | 0.66 | 0.69 | 0.62* | 0.59* | 0.63 | 0.66 | 0.58 | 0.63 |
| Passage Utility | **0.76** | **0.85** | **0.69** | **0.78** | **0.86** | **0.79** | 0.79 | 0.73* | 0.82 | **0.69** | **0.80** | **0.85** | **0.78** | 0.78 |

| | Llama-3.1-8B | | | | | | | Mistral-7B-v0.3 | | | | | | |
|---|---|---|---|---|---|---|---|---|---|---|---|---|---|---|
| | NQ | TQA | WebQ | SQuAD | PopQA | RefuNQ | AVG | NQ | TQA | WebQ | SQuAD | PopQA | RefuNQ | AVG |
| PPL | 0.75 | 0.80 | 0.68 | 0.74 | 0.83 | 0.60 | 0.73 | 0.63 | 0.71 | 0.57 | 0.65 | 0.64 | 0.62 | 0.64 |
| p(true) | **0.79** | **0.88** | **0.74** | 0.77 | 0.85 | 0.67* | 0.78 | 0.73 | 0.82 | **0.68** | 0.74* | 0.75* | 0.68* | 0.73 |
| Regular Entropy | 0.76* | 0.81 | 0.71* | 0.78 | 0.83* | 0.65 | 0.76 | 0.64 | 0.75 | 0.62* | 0.65 | 0.66 | 0.60 | 0.65 |
| Semantic Entropy | 0.72 | 0.82* | 0.66 | 0.78* | 0.81 | 0.59 | 0.73 | 0.66* | 0.78* | 0.66 | 0.73 | 0.74 | 0.61 | 0.70 |
| Passage Utility | 0.77 | 0.82 | 0.72 | **0.83** | **0.87** | **0.81** | 0.80 | **0.74** | **0.83** | **0.68** | **0.82** | **0.85** | **0.80** | 0.79 |

Table 1: AUROC values for QA models based on Gemma2-9B/27B, Llama-3.1-8B, and Mistral-7B-v0.3 on Natural Questions (NQ), TriviaQA (TQA), WebQuestions (WebQ), SQuAD, PopQA, and RefuNQ test sets. The best values (per model and dataset) are highlighted in **bold**; we also mark with * next best values which are significantly different using the paired De Long test ($p < 0.05$). For example, on TQA with Gemma2-9B, p(true), the second best performing is significantly different from the Passage Utility which performs best and by extension models with lesser values than p(true) are also significantly different.

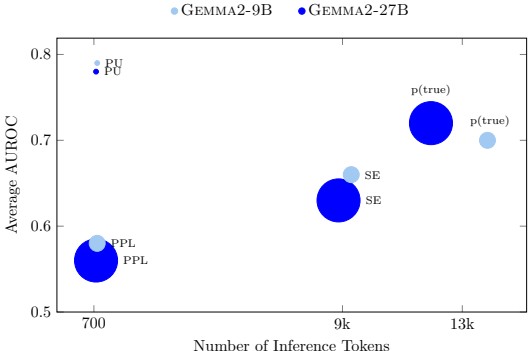

Figure 2: Average AUROC across our six development sets (y-axis) with respect to number of input tokens at inference time (x-axis) and number of parameters (size of the circles). We present results for perplexity (PPL), Semantic Entropy (SE), p(true), and Passage Utility (PU). We exclude Entropy (which is close to SE) for readability. We compare the smaller Gemma2-9B and its bigger version 27B. Thinner circles positioned in the left corner are better.

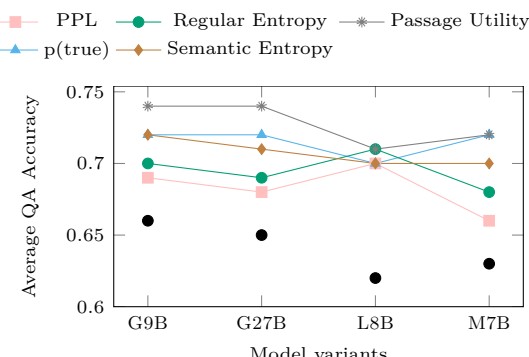

Figure 3: Average accuracy (across our six test sets) with Gemma2 sizes 9B and 27B (G9B, G27B), Llama-3.1-8B (L8B), and Mistral-7B-v0.3 (M7). Black dots: QA models always answer; colour dots: QA models answer 80% of the cases they are most confident about.

It is important to note that our approach is lite-weight at inference time. In Figure 2 we report average AUROC per method with respect to the number of input tokens at inference time and the number of parameters involved in uncertainty estimation. We report scores for Gemma2-9B and its bigger version Gemma2-27B. As can be seen, our approach which is based on a BERT encoder (Devlin et al., 2019) with 110M parameters and a small number of input tokens achieves on aggregate better performance than more expensive approaches. For the 27B QA model, p(true) edges closer to passage utility, however, at the expense of thousands of input tokens and ∼26 billions more parameters. This indicates that p(true) will be less efficient in QA settings where latency and cost are critical. In Appendix B we provide a cost analysis in terms of model calls.

**Passage Utility leads to selective answering.** Model uncertainty can be used to decide whether to provide an answer to a question or not. Figure 3 shows average accuracy when the target QA models choose to answer 80% of the cases they are most confident about. For comparison, we also show QA accuracy when always answering, i.e., black bold dots. All uncertainty quantification approaches improve model accuracy. Across different LLM QA backbones, Passage Utility performs on par with or better than more expensive uncertainty estimation approaches such as p(true). For instance, when looking at selective performance according to Passage Utility, the biggest Gemma2-27B model improves by +9 points (0.74 vs 0.65). In Appendix G.3 we report the full set of selective accuracies at different thresholds.

**Passage Utility shows potential for retrieval reranking.** To further assess the quality of the Passage Utility scores and to highlight their potential for retrieval reranking we carry out the following ablation study. Since most previous work (Asai et al., 2024; Xie et al., 2024; Yoran et al., 2024) on retrieval augmented QA prompts the QA model with the top-5 (or less) input passages, we hypothesize that our passage utility score could be an effective reranking method after retrieval (Nogueira et al., 2019; Ma et al., 2024; Yao et al., 2024). We test this hypothesis by computing the accuracy of the Gemma2-9B QA model when prompted with the top-$k$ passages (with $k$ in the range of $\{5, 3, 1\}$) out of a sample of $|R| = 10$ passages provided by an external retriever.

In Figure 4, we compare performance under two passage-ranking strategies: one based on relevance scores from an external retriever (gray), and the other based on the QA model's self-assessed utility of individual passages, following Yao et al. (2024). We report two self-assessment variants, one using the predictions of the Passage Utility model (red) and another one based on the perplexity of the QA model when answering with individual retrieved passages (blue). Passages that yield answers with lower perplexity should be ranked first. Figure 4 shows average accuracy values across five development sets (NQ, TQA, WebQ, SQuAD, and PopQA), at different cutoff values ($k$). As can be seen, the QA model achieves higher accuracy when passages are ranked according to their utility. This finding suggests that Passage Utility scores indeed reflect which passages are useful for the target QA model.

**Passage Utility struggles with multi-hop questions.** Our approach estimates uncertainty by predicting individual passage utility and selecting the maximum utility score from a set of retrieved passages. A potential limitation of this method emerges in multi-hop questions that require evidence from multiple passages (Yang et al., 2018; Pal et al., 2022). Specifically, the QA model is unable to answer correctly when prompted with any individual passage (resulting in uniformly low utility scores), yet succeeds when given the complete set of passages (in which case the maximum utility score over individual passages fails to reflect this outcome). To quantify this limitation, we evaluate our approach on the widely used HotPotQA dataset (Yang et al., 2018), using the splits as provided by Trivedi et al. (2023), and the same retrieval settings as defined above (Section 4). Table 2 shows AUROC values for all uncertainty estimation methods. In Appendix E, we provide an analysis on a small set of multi-hop questions making use of synthetic HotPotQA data.

Across models, our approach is on par with p(true) and sequence entropy, and better than perplexity (Table 2). Manual inspection of 100 examples from the development set with Gemma2-9B as the target QA model, reveals two major trends. Firstly, the QA model frequently (48 cases) manages to correctly answer multi-hop questions using only one of the required 'hop' passages, often not needing the entire set. This observation aligns with recent findings in Joren et al. (2025). Secondly, in numerous instances (29 cases), the retrieved passages did not contain any useful evidence, leading the model to answer incorrectly even when prompted with the full set. This underscores the inherent difficulty of effective retrieval for complex questions. Many studies (Jeong et al., 2024; Trivedi et al., 2023; Lin et al., 2025) tackle this challenge through sophisticated QA pipelines. Notably, certain approaches decompose complex questions into sub-questions that can be answered independently. In such multi-hop pipelines, our approach could be naturally applied at the sub-question level.

## 6 Discussion

**Key Properties and Usability Scenarios** Each uncertainty quantification approach comes with its own advantages and limitations. This entails that the choice of a specific method depends on criteria like available resources, desired latency, and the necessary level of control and trustworthiness for the QA system. For

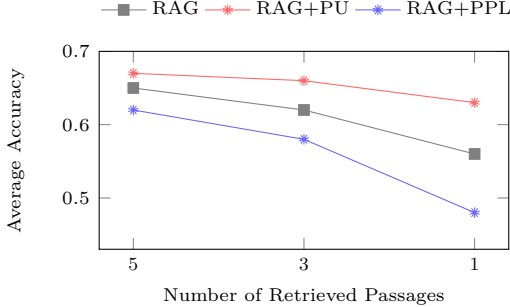

Figure 4: Average RAG accuracy with Gemma2-9B across the five QA development sets. Points on the x-axis correspond to different context sizes, when taking the top-$k$, passages according to query relevance (gray) and self-assessment via perplexity (PPL; blue), and Passage Utility (PU; red).

|  | G9B | G27B | L8B | M7B |
|---|---|---|---|---|
| PPL | 0.61 | 0.59 | 0.73 | 0.70 |
| p(true) | 0.75* | **0.78** | **0.81** | **0.78** |
| Regular Entropy | 0.64 | 0.60 | 0.77* | 0.71 |
| Semantic Entropy | 0.70 | 0.65* | 0.74 | 0.73 |
| Passage Utility | **0.78** | 0.77 | 0.77* | 0.74* |

Table 2: AUROC values for QA models based on Gemma2-9B/27B, Llama-3.1-8B, and Mistral-7B-v0.3 on HotPotQA test set. Best values per model are highlighted in **bold**; we also mark with* next best values which are significantly different using the paired De Long test ($p < 0.05$).

|  | Low latency | No Training Data | No Fine-tuning | Recognises Erroneous Evidence | Mitigates over-confidence |
|---|---|---|---|---|---|
| PPL | ✓ | ✓ | ✓ |  |  |
| p(true) |  |  | ✓ | ✓ | ✓ |
| Semantic Entropy |  | ✓ | ✓ |  |  |
| Passage Utility | ✓ |  |  | ✓ | ✓ |

Table 3: Categorization of the uncertainty estimation approaches studied in Section 5 according to different properties (table headers).

example, in high-stakes applications, a method that favours higher rates of false positives (thereby allowing human intervention) and reduces the chance of overconfident false negatives would be better, even if it requires training data and regular updates.

In Table 3, we summarise existing features (and limitations) of various uncertainty estimation approaches as we compare them to our work. The first column reports latency, as previously discussed in Figure 2. Retrieval augmented QA models can err as a result of being exposed to erroneous sources such as misleading passages (Xie et al., 2024) or inaccurate training data (Vu et al., 2024). Information-theoretic methods are not equipped with an explicit mechanism to deal with these cases (Farquhar et al., 2024; Soudani et al., 2025). Moreover, these methods are also known to suffer from over-confidence (Simhi et al., 2025; Soudani et al., 2025; Sung et al., 2025). While p(true) may be able to detect these challenging cases to a certain extent, Passage Utility can be *specifically* trained to recognise them. In terms of supervision, both p(true) and Passage Utility require task-specific training examples, and the performance of both approaches deteriorates on out-of-distribution examples (Table 8).

**Training Data Requirements** Our approach requires question-answer pairs to curate a dataset with retrieved passages and utility scores for training. However, general or task specific training datasets could be generated semi-automatically Li & Zhang (2024); Wei et al. (2024). Moreover, in experiments we show that in some QA tasks such as WebQ or PopQA this training data can be relatively small, i.e., in the region of 2.5k or 10k respectively.

**Fine-Tuning Requirements** Our approach, by design, requires fine-tuning to adapt to new QA tasks or models, as its core aim is to model the accuracy behavior of the target QA model. To enhance versatility across QA tasks, a unified training set encompassing diverse QA tasks could be compiled to train a single passage utility predictor. More practically, advanced training schemes promoting generalization, such as

meta-learning with a varied set of QA tasks and QA model examples, could be employed to develop a single passage utility predictor.

In addition, while the utility predictor may necessitate recalibration for distinct QA models or tasks, its performance on out-of-distribution scenarios (Appendix D) establishes it as a robust warm-up base model. This allows for subsequent fine-tuning with a reduced number of target examples (Kamath et al., 2020; Zhang et al., 2021). Finally, as the utility predictor relies on a small model, the cost of fine-tuning in terms of resources and time is low.

## 7 Conclusions

In this work we address uncertainty estimation in the context of retrieval augmented QA with a method that relies on individual passage utilities. Key in our approach is the definition of utility in terms of the behaviour of the QA model and whether it is able to provide a correct answer given a retrieved passage. We train a small neural model on passage utility judgements elicited from the QA model's responses and use utility predictions to estimate answer uncertainty. Experimental results show that our uncertainty estimator is competitive or better than existing strong methods while being light-weight. Future work could extend this approach to long-form generation tasks (Stelmakh et al., 2022; Gao et al., 2023; Min et al., 2023; Zhang et al., 2024a) where evaluating whether answer correctness is more challenging (Zhang et al., 2024b) and to multi-modal QA scenarios (Borszukovszki et al., 2025).

## Acknowledgments

We thank the action editor and anonymous reviewers for their constructive feedback. We want to thank Ivan Titov and Gustavo Giménez for useful discussions; and Alex Yu for sharing the synthetic multi-hop QA scripts. We are also grateful to Harry Sfyrakis and the UK Research and Innovation via the Innovate UK award (10092998) to Algomo Limited. This work was supported by the UK Engineering and Physical Sciences Research Council (grant EP/W002876/1). We also gratefully acknowledge the support of the Edinburgh International Data Facility (EIDF) and the Data-Driven Innovation Programme at the University of Edinburgh.

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

## A Ablation of the Passage Utility Predictor Training Objective

Table 4 shows AUROC results on answer uncertainty prediction with Passage Utility estimators trained with different variants of the objective in Equation (5). The first row shows the full objective (see training details in Section F.2), the second row shows a variant where the ranking objective uses only entailment utility annotations ($e$), and in the third row the objective is solely based on accuracy prediction ($\mathcal{L}_{BCE}$). As can be seen, there is a drop in performance when the pairwise ranking loss is not used (i.e., last line of Table 4); this component of the objective provides a smoother signal on passage utility, which is empirically beneficial. However, when the pairwise ranking loss is only based on entailment, performance drops by several points, highlighting the importance of combining both to model QA answering behaviour.

Table 5 reports various ablation studies (with GEMMA2-9B on the WebQ development set) with different instantiations of the Passage Utility score $v_{\mathcal{M}}$ and our training objective. We present different ways to combine accuracy ($a$) and entailment ($e$) scores to induce utility rankings with the pairwise ranking loss $\mathcal{L}_{rank}, (\cdot)$ (Equation 3). These include average $(a + e)/2$ as in Equation 2, addition $(a + e)$, addition but inverting the entailment score $((e|1 - e) + a)$ when the passage yields an inaccurate answer ($a = 0$), or when one of the two is given a zero weight, i.e., ($a$) or ($e$) alone. We asses these ranking variants when using the ranking objective alone (rightmost block) as well as when combined with the binary cross-entropy objective $\mathcal{L}_{BCE}$ (Equation 5) (middle block); we also report performance when only the $\mathcal{L}_{BCE}$ objective is

| Objective Terms | G9B | L8B | M7B |
|---|---|---|---|
| $\mathcal{L}_{rank}, (e+a)/2 + \lambda \mathcal{L}_{BCE}$ | 0.80 | 0.81 | 0.80 |
| $\mathcal{L}_{rank}, (e) + \lambda \mathcal{L}_{BCE}$ | 0.73 | 0.73 | 0.71 |
| $\mathcal{L}_{BCE}$ | 0.78 | 0.78 | 0.80 |

Table 4: Answer uncertainty estimation with Passage Utility predictors trained with different variants of the objective in Equation (5). AUROC values for GEMMA2-9B (G9B), LLAMA3.1-8B (L8B), and MISTRAL-7B-V0.3 (M7B) are averaged over development sets.

| $\mathcal{L}_{BCE}$ | $\mathcal{L}_{rank} + \lambda \mathcal{L}_{BCE}$ | | | | $\mathcal{L}_{rank}$ | | | | |
|---|---|---|---|---|---|---|---|---|---|
| | $(e+a)/2$ | $(e+a)$ | $((e\vert 1-e)+a)$ | $(e)$ | $(e+a)/2$ | $(e+a)$ | $((e\vert 1-e)+a)$ | $(e)$ | $(a)$ |
| 0.71 | 0.75 | 0.74 | 0.72 | 0.63 | 0.65 | 0.63 | 0.62 | 0.61 | 0.53 |

Table 5: Answer uncertainty estimation with Passage Utility predictors trained with different variants of the objective in Equation (2). AUROC values are for GEMMA2-9B and the WebQ development set.

used (leftmost block). All model variants are trained with rank-based model selection and $\lambda = 0.25$ following training details in Section F.2.

When looking at the pairwise ranking objective alone (rightmost block), entailment dominates the ranking and the utility scores learnt; $(e)$ pairwise ranking as well as $(a+e)$ variants yield utility scores that have similar discriminative power. The $(a)$ variant exhibits the worst performance, given that many pairs are discarded due to having the same utility. AUROC values improve when the pairwise ranking is combined with the cross-entropy objective (middle block). In this case, in addition to enforcing the pairwise ranking with $\mathcal{L}_{rank}, (\cdot)$, the utility scores are regularised. In other words, the utility scores of a pair of passages with the same accuracy (i.e., only ordered by entailment) will end up closer (differ less) than the utility scores of a pair of passages with different accuracies (differ more). This is reminiscent of the Bradley-Terry model (Bradley & Terry, 1952): in the first case, one passage is better than the other with a small probability, while in the second case the probability of one being better than the other is higher. The exception here is the $\mathcal{L}_{rank}, (e) + \lambda \mathcal{L}_{BCE}$ variant where the pairwise ranking by entailment may often contradict the binary cross-entropy signal.

## B  Test Time Cost of Uncertainty Estimation Methods

Table 6 shows the cost of estimating uncertainty for question $x$, measured by the number of inference calls required. Simple information theoretic methods (e.g., PPL) require a single call to the target QA model with the retrieval augmented QA prompt (i.e., $|R|$ retrieved passages and question $x$). However, approaches that estimate uncertainty based on diversity (e.g., Regular Entropy, Semantic Entropy, and p(true)) require generating $N$ answers, i.e., $N$ inference calls with the retrieval augmented QA prompt. In addition, Semantic Entropy requires the computation of answer clusters (i.e., grouping answers with the same meaning), so additional calls to an entailment model are required to compare the set of sampled answers. p(true) requires one additional LLM call to elicit a True/False answer but with a very long prompt including in-context examples and the assessment question with the $|R|$ retrieved passages, sampled and most likely answers, and question $x$ (see Table 15). In contrast, our approach requires $|R|$ utility predictions with a BERT-sized model. For instance, in our experimental setup with $N = 10$ samples and retrieval augmented QA with $|R| = 5$, the Semantic Entropy approach would require 11 forward passes with the QA model prompted with 5 passages (one the greedy candidate and 10 random samples) plus 45 calls to the entailment model. Our approach requires 5 forward passes with the BERT-based model.

| Methods | Inference Calls at Test Time |
|---|---|
| PPL | $1\,\mathrm{G}$ |
| p(true) | $(N+1)\,\mathrm{G} + 1\,\mathrm{E}$ |
| Regular Entropy | $(N+1)\,\mathrm{G}$ |
| Semantic Entropy | $(N+1)\,\mathrm{G} + \binom{N}{2}\,\mathrm{E}$ |
| Passage Utility | $|R|\,\mathrm{BERT\text{-}F}$ |

Table 6: Number and type of inference calls required to estimate answer uncertainty for question $x$ and set of retrieved passages $R$. G means inference is performed with a retrieval augmented QA model, i.e., an LLM forward pass with the prompt including the set of $|R|$ retrieved passages and question $x$ to generate a candidate answer $y$. E is inference with an evaluation model, e.g., a forward pass to ask an LLM for correctness in p(true) or a forward pass with an entailment model in Semantic Entropy. Bert-F is an inference call to predict passage utility for passages $p$ in $R$ and question $x$.

| | | G9B | G27B | L8B | M7B |
|---|---|---|---|---|---|
| RAcc↓IndAcc↑ | $a$ | 0.12 | 0.13 | 0.15 | 0.15 |
| | $a+e$ | 0.04 | 0.04 | 0.05 | 0.06 |
| RAcc↑ IndAcc↓ | $a$ | 0.01 | 0.01 | 0.01 | 0.01 |

Table 7: Average proportion of cases (five development sets) where at least one individual passage in $R$ leads a correct answer (IndAcc↑) but the QA model prompted with $R$ yields an incorrect one (RAcc↓) (and vice versa RAcc↑ IndAcc↓).

## C  Generality of the Uncertainty Aggregation Strategy

To validate the intuition behind our uncertainty estimation step (Section 3.2), we compare the behaviour of the QA model when prompted with individual passages in $R$ versus when prompted with $R$ (i.e., the top-$|R|$ passages). In particular, we want to inspect the proportion of cases where our strategy of taking the maximum utility score among the passages in $R$ does not agree with the entire set accuracy. In other words, we are interested in cases where the QA model is accurate with at least one individual passage (IndAcc↑) but answers incorrectly when prompted with $R$ and vice versa. In this study, we consider two individual passage accuracy variants (related to the combined definition of Passage Utility Section 3.1). One is based on accuracy $a$ (see Section 4) and is either 0 or 1. The other one is also based on accuracy but smoothed by entailment $e$ (computed by an off-the-shelf entailment model; see Section 4) and downgrades cases where $a=1$ into $a=0$ if $e < 0.5$. The latter occurs in cases where the QA model produces an answer that is accurate but not entailed by the passage.

We analyse the behaviour of four QA models (LLMs) across five datasets in our experimental setup. Table 7 shows the proportion of model disagreements across development sets. We can see that such disagreements amount to a relatively small number in both settings, i.e., when at least one individual passage in $R$ yields a correct answer (IndAcc↑) but the QA model prompted with $R$ yields an incorrect answer (RAcc↓) and vice versa (RAcc↑ IndAcc↓). The results in Table 7 confirm that our aggregation approach based on individual passages is fairly general. It approximates answer uncertainty when prompting with $|R|$ passages while avoiding the complexity of estimating uncertainty over all possible combinations of input passages in terms of number and order.

Our study is related to the issue of understanding LLM sensitivity to external evidence (Xie et al., 2024; Liu et al., 2024b), i.e., how the type of evidence (supportive, contradictory, irrelevant, or misleading), the amount, and order of presentation affect LLM predictions and interact with parametric knowledge. The Passage Utility predictor is trained to predict the error of a target QA model (LLM) on answering questions, independent of the type of passage or any memorized knowledge. Given a question-passage pair, if the LLM relies on its memorized knowledge rather than adapting to the passage and still produces the correct answer, or conversely, adapts to the passage but produces an incorrect answer, then the Passage Utility predictor should reflect this outcome by predicting a correct or incorrect answer accordingly Xie et al. (2024).

Note that the Passage Utility predictor is meant to be synchronized with the target QA model and judgments of what is (or not) a correct answer. If an answer to a question changes, the target QA model answer correctness on this question may also change, and the Passage Utility predictor should also reflect this (i.e., it should be adapted with new examples). Multiple passage interactions studied in datasets with synthetic evidence (Longpre et al., 2021; Xie et al., 2024) are observed to a lesser extent in our experiments with six datasets and external retrievers. This has also been recently pointed out in Hagström et al. (2025). Our approach could be combined with additional features to capture more complex interactions (Dong et al., 2018). Investigating and understanding the relation between QA model uncertainty and (improving) context utilization is an interesting topic on its own right (Xie et al., 2024; Longpre et al., 2021; Hagström et al., 2025) but out of scope for this paper.

## D  Out-of-distribution Generalization of Uncertainty Estimation

We assess the generalization ability of our Passage Utility estimator both in terms of new QA tasks and QA models. We train a unique Passage Utility predictor for the GEMMA2-9B model. Following previous work on question answering and out-of-distribution (o.o.d) scenarios (Kamath et al., 2020; Zhang et al., 2021), we train it on the SQuAD dataset and then use it to predict zero-shot (i.e., without further fine-tuning) passage utilities on all other datasets (test set). As p(true) relies on 20 in context training examples, we also evaluate its ability to generalise in out-of-distribution settings.

Table 8 (QA Task block) shows AUROC for answer uncertainty estimation in o.o.d scenarios. As an upper bound, the i.i.d block of the table shows AUROC values in the in-distribution scenario for Passage Utility and p(true). We compare o.o.d performance w.r.t. PPL and Semantic Entropy which do not rely on training examples. Although Passage Utility performance decreases in o.o.d settings, it remains competitive in four out of five datasets. In these four cases, it is always statistically significantly different from the PPL method and comparable to p(true) and Semantic Entropy. Interestingly, p(true)'s performance also drops in all o.o.d test sets showing that relying on a fixed number of in-context learning examples is neither a robust nor scalable adaptation method.

To understand the observed performance drop, we conducted a comparative analysis of passage utility predictions for 50 question-passage pairs. We examined predictions from a utility predictor trained on WebQ data (i.i.d) versus one trained on SQuAD (o.o.d). We sampled 10 WebQ test questions: 5 for which the predicted utility decreased from high under i.i.d. conditions to low under o.o.d. conditions (changing from true negative to false positive), and 5 for which it increased in the opposite direction (changing from true positive to false negative). In both scenarios, the o.o.d predictions were predominantly influenced by token overlap and similarity. We hypothesize that when faced with o.o.d questions (e.g., in terms of type, length, or topic), the prediction mechanism defaults to predictions based on a general notion of question-passage similarity, disregarding whether the QA model can answer the question with the given passages. For passages with low (high) similarity to the question, it will predict low (high) utility scores. This behaviour may be a consequence of under-training, suggesting that the Passage Utility predictor cannot predict the accuracy of the target QA model on these o.o.d questions and their corresponding retrieved passages.

To empirically validate these observations, we calculate Spearman's $\rho$ correlation coefficients between the predicted Passage Utility scores and the corresponding retriever relevance scores (as computed by Contriever-MSMARCO; Izacard et al. 2022) for each passage, examining correlations separately for both i.i.d and o.o.d predictions. We use the Retriever Score because it measures the semantic relation between questions and retrieved passages. In agreement with our manual inspection, there is a positive correlation between Passage Utility and Retriever Score in the o.o.d setting (0.536 with $p$-value $< 0.01$), suggesting that Passage Utility falls back to textual similarity. In contrast, the i.i.d setting exhibits a negative and weaker correlation (-0.318 with $p$-value $< 0.05$), indicating that training successfully aligns Passage Utility predictions with the target QA model's *actual* performance rather than superficial retriever similarity scores.

We also evaluate generalisation to a new QA model, by training the Passage Utility predictor on utility labels observed for GEMMA2-9B and use its predictions to estimate uncertainty for the bigger GEMMA2-27B model. We also evaluate p(true) in this o.o.d setting, i.e., we create p(true) prompts as usual with GEMMA2-27B generated answers (greedy and sampling), but ask GEMMA2-9B to judge the probability of the most likely

| | | GEMMA2-9B | NQ | TQA | WebQ | PopQA | RefuNQ |
|---|---|---|---|---|---|---|---|
| **QA Task** | i.i.d | p(true) | 0.73 | 0.75 | 0.67 | 0.81 | — |
| | | Passage Utility | 0.76 | 0.85 | 0.69 | 0.86 | — |
| | o.o.d | PPL | 0.64 | 0.68$^*$ | 0.52 | 0.59$^*$ | 0.51 |
| | | Semantic Entropy | **0.70** | 0.73 | 0.58$^*$ | **0.73** | 0.59$^*$ |
| | | p(true) (SQuAD) | 0.67 | 0.63 | **0.63** | 0.72 | 0.62 |
| | | Passage Utility (SQuAD) | 0.65 | **0.79** | 0.60 | 0.72 | **0.79** |
| | | GEMMA2-27B | NQ | TQA | WebQ | PopQA | RefuNQ |
| **QA Model** | i.i.d | p(true) | 0.77 | 0.83 | 0.67 | 0.79 | 0.60 |
| | | Passage Utility | 0.73 | 0.82 | 0.69 | 0.87 | 0.80 |
| | o.o.d | PPL | 0.64$^*$ | 0.50 | 0.53 | 0.53 | 0.51 |
| | | Semantic Entropy | 0.68 | 0.62 | 0.59 | 0.66 | 0.58 |
| | | p(true) (GEMMA2-9B) | 0.73 | 0.71$^*$ | 0.64$^*$ | 0.75$^*$ | 0.59$^*$ |
| | | Passage Utility (GEMMA2-9B) | **0.75** | **0.80** | **0.68** | **0.87** | **0.79** |

Table 8: Out-of-domain performance of Passage Utility predictor for GEMMA2-9B both in terms of QA task and QA model. i.i.d blocks report AUROC values from our main in-distribution experiments (Table 1); o.o.d blocks contain the o.o.d comparison. Best values are highlighted in **bold**; we also mark with $^*$ next best values which are significantly different using the paired De Long test ($p < 0.05$). For the QA task, Passage Utility and p(true) are supervised with SQuAD data and evaluated on NQ, TQA, WebQ, PopQA, and RefuNQ test data. For the QA model, Passage Utility predictors are trained on GEMMA2-9B and used to estimate uncertainty for GEMMA2-27B; p(true) assessment is provided by GEMMA2-9B.

answer being true. Table 8 (QA Model block) shows AUROC results; for reference, we include in-distribution values (i.i.d block) and compare the o.o.d results with comparison methods. Passage Utility outperforms all other methods across the board, and while still competitive, p(true) exhibits a higher decrease in performance. These preliminary results suggest that in the context of retrieval augmented QA, models behave alike (also suggested by the distribution of observed correct/incorrect individual passage utilities in Table 12). This highlights practical benefits of our approach, such as training a base Passage Utility predictor using data generated by a less expensive model or developing a more general predictor applicable across multiple QA models.

Finally, we train a unique Passage Utility predictor for *all* QA tasks and assess its generalisation capabilities. To this end, we train a predictor for the GEMMA2-9B model on a random sample drawn from the five training sets of size equivalent to the training sets of the individual QA task predictors. Specifically, we took 10k from NQ, TQA, and SQuAD, 3k from PopQA, all WebQ, totalling 35,474 instances for training and 500 instances from each for validation; the number of pairwise instances in the final curated training set is 354,379. For p(true), we mixed 4 randomly sampled examples from each dataset as in-context training examples. We follow the same training procedure as described in Section F.2 (with combined model selection and $\lambda = 1$). We show AUROC results in Table 9. Across the six test sets, the unique Passage Utility predictor trained on a mix of QA tasks (bottom block) achieves similar performance to the individual predictors trained on per-task datasets (top block). The unique predictor keeps comparable or better performance than p(true) and outperforms PPL and Semantic Entropy. This preliminary study suggests that is feasible to train a more general predictor for various QA tasks.

# E   Synthetic Qualitative Analysis of Passage Utility in Multi-hop QA

In multi-hop QA Passage Utility may fail at cases where the QA model cannot answer when prompted with any individual 'hop' passage, but answers correctly when prompted with the full set. To analyse this limitation, we carry out a qualitative analysis. As discussed in Section 5, there are two scenarios in retrieval augmented multi-hop QA (given existing datasets and current LLMs). Either models can answer with one passage (thus no multi-hop) or retrieval completely fails. Thus, to be able to pinpoint actual multi-hop QA

|  |  | NQ | TQA | WebQ | SQuAD | PopQA | RefuNQ |
|---|---|---|---|---|---|---|---|
| Task | p(true) | 0.73 | 0.75 | 0.67 | 0.68 | 0.81 | 0.62 |
|  | Passage Utility | 0.76 | 0.85 | 0.69 | 0.80 | 0.86 | 0.79 |
| Unique | PPL | 0.64$^*$ | 0.68 | 0.52 | 0.59 | 0.59 | 0.51 |
|  | Semantic Entropy | 0.70 | 0.73$^*$ | 0.58$^*$ | 0.63$^*$ | 0.73 | 0.59 |
|  | p(true) | **0.71** | 0.70 | **0.68** | 0.63$^*$ | 0.75$^*$ | 0.66$^*$ |
|  | Passage Utility | 0.69 | **0.83** | **0.68** | **0.79** | **0.85** | **0.79** |

Table 9: Generalisation of a Unique Passage Utility predictor for Gemma2-9B trained on a mix of QA tasks and evaluated on the six test sets (lower block). Best values are highlighted in **bold**; we also mark with $^*$ next best values which are significantly different using the paired De Long test ($p < 0.05$). The upper block reports AUROC values for the Passage Utility from predictors trained for Gemma2-9B on indiviual QA tasks' training datasets. We also compare p(true) when in-context learning examples are all from the same QA task (upper block) versus from a mix of tasks (lower block).

cases and expose the limitations of our approach, we make use of synthetic HotPotQA data (Yang et al., 2018).

First, we use the sets of gold passages provided in the HotPotQA dataset to categorise question types as follows. We prompt the QA model with individual gold passages and with combinations thereof, measuring accuracy. Out of the 500 test questions, 475 are answerable with only one gold 'hop' passage, 13 require multiple passages, and 12 are unanswerable. Secondly, we focus on the 13 requiring multi-hop questions and insert gold 'hop' passages within the retrieved set for those where retrieval completely fails. Then, we run QA and uncertainty estimation. In 6 of 13 cases, our approach fails to predict that the QA model's answer will be correct. It is important to note, however, that these 13 cases represent only 2.6% of the total (13 out of 500).

# F   Experimental Details

## F.1   Datasets and Splits

In our experiments, we use six QA tasks which we describe below. Table 10 shows dataset statistics and example question-answers pairs.

**Natural Questions**   (NQ; Kwiatkowski et al. 2019) is a QA dataset compiled from real user questions submitted to the Google search engine. As part of the dataset curation process, annotators judge the quality of questions and associate them with a short answer that can be extracted from a related Wikipedia page.

**TriviaQA**   (TQA; Joshi et al. 2017) is a question answering dataset designed for training and evaluating machine learning models on open-domain question answering tasks. The dataset was created by gathering questions from trivia websites, along with their corresponding answers, to provide a broad range of factual questions.

**WebQuestions**   (WebQ; Berant et al. 2013) was mined off questions generated with the Google Suggest API. The answers to the questions are defined as Freebase entities (i.e., their string label) and were elicited by Amazon Mechanical Turk (AMT) annotators.

**SQuAD**   (Rajpurkar et al., 2016) contains questions formulated by AMT annotators based on a given Wikipedia paragraph, with the answer being a short span in that paragraph. Annotators were encouraged to use paraphrasing when writing the question. The answer types not only cover named entities but also other categories such as noun- and verb-phrases.

**PopQA**   Mallen et al. (2023) is an open-domain QA dataset, focusing on popular culture topics, such as movies, TV shows, music, and sports. It contains question-answer pairs derived from (subject, relation,

| Dataset | Train | Dev | Test | Example Question | Example Answer |
|---|---|---|---|---|---|
| NQ | 79,168 | 8,757 | 3,610 | Who plays Letty in Bring it on all or nothing? | Francia Raisa |
| TQA | 78,785 | 8,837 | 11,313 | Who was the first artistic director of the National Theatre in London? | Lord Laurence Olivier |
| WebQ | 2,474 | 361 | 2,032 | What party was Andrew Jackson? | Democratic-Republican Party |
| SQuAD | 78,713 | 8,886 | 10,570 | What is the Grotto at Notre Dame? | A Marian place of prayer and reflection |
| PopQA | 10,000 | 1,267 | 3,000 | Who was the director of Champion? | Rabi Kinagi |
| RefuNQ | — | — | 2,173 | Who does the voice over in the Requirtion? | — |

Table 10: Dataset statistics, number of instances per Train/Development(Dev)/Test sets, and example question-answer pairs (all taken from the Dev set except for RefuNQ).

object) triples in Wikidata . Triples were translated into natural language and the object entity was taken to be the gold answer. The collection process focused on gathering questions about subject entities of varying popularity.

**RefuNQ**   Liu et al. (2024a) is derived from NQ and consists of answerable and unanswerable questions. Unanswerable questions are created by replacing entities in the original NQ question by non-existing concepts.

We follow previous work (Lee et al., 2019) and use only the question and gold answers, i.e., the open versions of NQ, TQA, and SQuAD. We use the unfiltered TQA dataset. We follow the train/dev/test splits as used in previous work Lee et al. (2019) and randomly split PopQA. RefuNQ only provides a test set so our experiments on this dataset are zero-shot from a Passage Utility predictor trained on SQuAD. We follow Farquhar et al. (2024) and use 400 test examples randomly sampled from the original larger test datasets for evaluation of uncertainty quantification.

## F.2   Implementation Details

**QA Models**   For all question answering tasks, we use the off-the-shelf Contriever-MSMARCO tool (Izacard et al., 2022) to retrieve sets of passages $R$ for question $x$ from Wikipedia and the official Wikipedia embeddings based (2018 snapshot) as our document knowledge-base. For PopQA, we follow the work of Mallen et al. (2023) who also use the full 2018 English Wikipedia dump.

The QA prompt used for all models (embedded in the corresponding chat templates) is shown in Table 14. For inference, we set the maximum number of generated tokens to 50 for both the greedy (most likely answer) as well as temperature scaled (sampled candidates) decoding. We use vLLM for inference (Kwon et al., 2023). For all models, inference was run on a single A100-80GB GPU.

**Curated Passage Utility Dataset**   We train our passage utility predictor on a dataset $D_{\mathcal{M}}$ curated from benchmark $D$, e.g., WebQ, consisting of question and gold answer pairs $(x, y)$. For each question we retrieve the top-$k$ passages. Then, we pair question $x$ and retrieved passages $p$ with utility scores $v_{\mathcal{M}}$ which we obtain after running the QA model $\mathcal{M}$ on inputs $(x, p)$ and computing the generated answer accuracy and entailment scores (Section 3.1), i.e., we create tuples $(x, p, v_{\mathcal{M}})$. From the set of $k$ tuples for question $x$, we derive $\binom{k}{2}$ instances for our pairwise ranking loss.

In experiments, we use $k = 5$ retrieved passages per question. Table 11 reports the size (number of training instances) of the curated datasets for each QA task and model. From each question and set of top-5 retrieved passages, we derive 10 pairwise ranking instances, discarding those that have equal utilities (e.g., from the WebQ training split with 2,474 question-answer pairs, we curate 24,720 instances with MISTRAL-7B-V0.3). As our top-5 passages are obtained via a real retrieval module, i.e., not synthetically assembled, there are questions for which all passages in the top-5 set lead to a correct (incorrect) answer. In these cases, the

| Models | NQ | TQA | WebQ | SQuAD | PopQA |
|---|---|---|---|---|---|
| Gemma2-9B | 395,438 | 393,475 | 24,721 | 393,285 | 99,770 |
| Gemma2-27B | 395,426 | 393,477 | 24,723 | 393,293 | 99,778 |
| Llama-3.1-8B | 790,862 | 393,465 | 24,713 | 393,288 | 99,787 |
| Mistral-7B-v0.3 | 395,397 | 393,474 | 24,720 | 393,271 | 99,772 |

Table 11: Number of pairwise training instances in the curated datasets to train the Passage Utility predictor with the combined pairwise ranking and binary cross-entropy losses (Section 5).

| | Incorrect | Mixed | Correct |
|---|---|---|---|
| Gemma2-9B | 21% | 52% | 27% |
| Gemma2-27B | 20% | 53% | 27% |
| Llama-3.1-8B | 22% | 55% | 23% |
| Mistral-7B-v0.3 | 21% | 56% | 23% |

Table 12: Number of training instances in curated datasets to train the passage utility predictor with the combined pairwise ranking and binary cross-entropy losses (Section 5).

pairwise ranking is dominated by the entailment score (i.e., accuracy is the same). Table 12 shows the distribution of questions with all retrieved passages leading to the same accuracy (Correct/Incorrect) or mixed (Mixed) accuracies in the curated dataset for each QA task and model.

**Passage Utility Predictor Training Details** We train a different predictor for each target QA model and QA task. Given the large number of predictors required in our experiments, we initially tested the hyper-parameters used in Fang et al. (2024) on the NQ dataset and choose a set thereof for all predictor instances. We train each predictor for 3 epochs, with a batch size of 32 examples, learning rate equal to $2^{e-5}$, and weight decay 0.001 (with the exception of Llama-3.1-8B and WebQ where we used 0.01). For each predictor we performed search on values for $\lambda$, i.e., the contribution of the $\mathcal{L}_{BCE}$ loss (Equation 5), and different criteria for model selection, i.e., the best at pairwise ranking or at both pairwise ranking and accuracy prediction (combined).

Table 13 shows the configuration for each predictor. Table cells show selection criteria (R for ranking and C for combined) and the value for $\lambda$. At inference time we predict a single Passage Utility score given by the selected best checkpoint. For all predictor instances (except for all WebQ and PopQA predictors and the predictor for Llama-3.1-8B and NQ), we use half of the available training data to speed up experiments. Training and inference was run on a single A100-40GB GPU; training ranges from 2 to 12 hours depending on the dataset.

**Comparison Approaches** In this section, we describe additional answer uncertainty estimation methods (for which we present supplementary results in Section G). Maximum Sequence Probability (MSP) is based on the probability of the most likely answer and is computed as:

$$\mathrm{MSP}(x, R, \mathcal{M}) = 1 - P(y|x, R; \mathcal{M}). \tag{12}$$

Note that, in contrast to $\mathrm{PPL}(x, R, \mathcal{M})$ reported in the main section of the paper, this metric is biased by answer length, i.e., identifying an answer to have low probability (low confidence) because of its length. Despite the fact that QA models are instructed to produce short answers, they do not always follow instructions. For this reason, we consider perplexity a more accurate metric. Indeed, answer length could indicate that the model is uncertain about the answer. Thus, we also estimate answer uncertainty from the Average Answer Length (AvgAnsLen) as the average number of words in the sampled answers. As seen in Section G.3, Table 20, MSP and AvgAnsLen perform similarly across the board.

| Models | NQ | TQA | WebQ | SQuAD | PopQA | HotPotQA |
|---|---|---|---|---|---|---|
| Gemma2-9B | C, 0.25 | C, 0.25 | R, 0.25 | C, 1 | C, 1 | C, 0.25 |
| Gemma2-27B | C, 0.25 | C, 1 | C, 1 | C, 0.25 | R, 0.25 | C, 1 |
| Llama-3.1-8B | C, 0.25 | C, 0.25 | C, 0.25 | C, 1 | C, 1 | C, 0.25 |
| Mistral-7B-v0.3 | R, 0.25 | C, 1 | C, 0.25 | C, 0.25 | C, 0.25 | C, 1 |

Table 13: This table shows the $\lambda$ value and selection criteria (R for pairwise ranking or C for combined pairwise ranking and accuracy prediction) for each Passage Utility predictor in our experiments.

We also report Cluster Assignment (CA) which is a variant of SE without answer probabilities where the probability of each generated meaning (i.e., a cluster) is approximated from the number of answers in the cluster. We found that in general CA estimations are very close to Semantic Entropy ones.

Another uncertainty estimation approach is the negative mean Point-wise Mutual Information (PMI; Takayama & Arase 2019) over tokens; i.e., it compares the probability of answer sequence $y$ given a prompt with question $x$ and passages $R$ w.r.t the probability given by $\mathcal{M}$ to $y$ without any context. Intuitively, the higher the point-wise mutual information, the more certain the QA model is on generating $y$ (i.e., the answer is related to or depends on $x$ and $R$). PMI is computed as:

$$\text{PMI}(x, R, \mathcal{M}) = -\frac{1}{|y|} \sum_{t=1}^{|y|} \log \frac{p(y_t | y_{1..t-1}, x, R; \mathcal{M})}{p(y_t | y_{1..t-1}; \mathcal{M})}. \tag{13}$$

We also report Retriever Score as a baseline for Passage Utility. Instead of using the predicted Passage Utility we use the original relevance score assigned by the external retriever (i.e., Contriever MS-MARCO).

We use the implementation provided in Farquhar et al. (2024) to compute Regular Entropy, Semantic Entropy, Cluster Assignment, and p(true). Note that we do not include the supervised baseline reported in Farquhar et al. (2024) as the authors show that it underperforms simple information-theoretic metrics and in addition only works for white-box models. Note that while AvgAnsLen and Retriever Score do not strictly provide scores in the $[0, 1]$ interval, the package that computes AUROC finds ranking decision thresholds.[4]

**Metrics** We use the implementation provided in Farquhar et al. (2024) to compute AUROC, Accuracy at X% of rejection, and AURAC metrics.

We use Qwen2-72B-Instruct (Yang et al., 2024) to obtain accuracy judgments (i.e., $A$ judge, Section 4); specifically, we use the Activation-aware Weight Quantization (Lin et al., 2024), version Qwen2-72B-Instruct-AWQ. We prompt the accuracy evaluator with the prompt proposed in Sun et al. (2024), as we found it to perform well on our datasets. The accuracy evaluation (AccLM) prompt is shown in Table 16. In a sample of 840 generated answers human and LLM-based judgment of correctness agreed 98% of the time (Sun et al., 2024).

### F.3 Prompts

The prompt we use for our QA models is shown in Table 14. Table 15 illustrates the prompt used for the p(true) baseline. Table 16 shows the prompt used for the LLM-based accuracy metric.

## G Additional Results

### G.1 Reference Retrieval Augmented QA Accuracy

Table 18 shows retrieval augmented QA performance for the five QA models on the development and test sets of our six tasks. We report accuracy based on token overlap (Acc) as computed in previous work, i.e.,

---

[4]https://scikit-learn.org/stable/modules/generated/sklearn.metrics.det_curve.html

Retrieval augmented QA prompt

Knowledge:
[1] passage
[2] passage
...
[|R|] passage

Answer the following question with a very short phrase.

Question: question

Table 14: Prompt designed as user turn for QA models.

p(true) prompt

Question: question
Brainstormed Answers: most likely answer
sampled answer 1
...
sampled answer N
Possible answer: most likely answer
Is the possible answer:
A) True
B) False
The possible answer is: correct choice

...

Knowledge:
[1] passage
[2] passage
...
[|R|] passage

Question: question
Brainstormed Answers: most likely answer
sampled answer 1
...
sampled answer $N$
Possible answer: most likely answer
Is the possible answer:
A) True
B) False
The possible answer is:

Table 15: Prompt used for p(true) approach. Items in blue are filled with in-context examples from the training set and the current example being evaluated. $N$ represents the number of sampled answers. The "sequence of in-context examples" prefix is a sequence of examples taken from the training split with the same question format but with the answer to *The possible answer is:* resolved.

whether the gold answer is contained in the generated answer (Mallen et al., 2023; Asai et al., 2024; Xie et al., 2024) and accuracy using an LLM as a judge (AccLM). Note that AccLM is much higher than Acc across the board, which highlights the importance of using a better accuracy metric, especially when the target QA models are not fine-tuned.

Accuracy evaluation prompt.

You need to check whether the prediction of a question-answering system to a question is correct. You should make the judgment based on a list of ground truth answers provided to you. Your response should be "correct" if the prediction is correct or "incorrect" if the prediction is wrong.

Question: Who authored The Taming of the Shrew (published in 2002)?
Ground truth: ["William Shakespeare", "Roma Gill"]
Prediction: W Shakespeare
Correctness: correct

Question: Who authored The Taming of the Shrew (published in 2002)?
Ground truth: ["William Shakespeare", "Roma Gill"]
Prediction: Roma Gill and W Shakespeare
Correctness: correct

Question: Who authored The Taming of the Shrew (published in 2002)?
Ground truth: ["William Shakespeare", "Roma Gill"]"
Prediction: Roma Shakespeare
Correctness: incorrect

Question: What country is Maharashtra Metro Rail Corporation Limited located in?
Ground truth: ["India"]
Prediction: Maharashtra
Correctness: incorrect

Question: What's the job of Song Kang-ho in Parasite (2019)?
Ground truth: ["actor"]
Prediction: He plays the role of Kim Ki-taek, the patriarch of the Kim family.
Correctness: correct

Question: Which era did Michael Oakeshott belong to?
Ground truth: ["20th-century philosophy"]
Prediction: 20th century."
Correctness: correct

Question: Edward Tise (known for Full Metal Jacket (1987)) is in what department?
Ground truth: ["sound department"]
Prediction: 2nd Infantry Division, United States Army
Correctness: incorrect

Question: What wine region is Finger Lakes AVA a part of?
Ground truth: ["New York wine"]
Prediction: Finger Lakes AVA
Correctness: incorrect

Question: question
Ground truth: gold answer
Prediction: generated answer
Correctness:

Table 16: Prompt used for LLM-based accuracy evaluation.

## G.2 Results on Individual Passage Utility Prediction

Beyond using Passage Utility to estimate uncertainty in retrieval augmented QA, we evaluate how it performs on its own. Table 17 shows AUROC scores when using Passage Utility to predict accuracy for individual passages. We evaluate on the same samples from the test sets in Section 5 and Table 1 but per passage. We also include a perplexity baseline (PPL). Overall, results in Table 17 follow a similar pattern as those in the QA setting with top-5 passages (Table 1). Across the board, Passage Utility demonstrates strong performance in predicting the usefulness of individual passages. Moreover, these results highlight that the quality of uncertainty estimation strongly depends on the quality of individual Passage Utility predictions.

| | Gemma2-9B | | | | | | | Gemma2-27B | | | | | | |
|---|---|---|---|---|---|---|---|---|---|---|---|---|---|---|
| | NQ | TQA | WebQ | SQuAD | PopQA | RefuNQ | AVG | NQ | TQA | WebQ | SQuAD | PopQA | RefuNQ | AVG |
| PPL | 0.64 | 0.62 | 0.53 | 0.56 | 0.45 | 0.50 | 0.55 | 0.61 | 0.61 | 0.53 | 0.57 | 0.52 | 0.54 | 0.56 |
| Passage Utility | 0.77 | 0.80 | 0.71 | 0.74 | 0.90 | 0.71 | 0.77 | 0.74 | 0.8 | 0.69 | 0.76 | 0.90 | 0.72 | 0.77 |

| | Llama-3.1-8B | | | | | | | Mistral-7B-v0.3 | | | | | | |
|---|---|---|---|---|---|---|---|---|---|---|---|---|---|---|
| | NQ | TQA | WebQ | SQuAD | PopQA | RefuNQ | AVG | NQ | TQA | WebQ | SQuAD | PopQA | RefuNQ | AVG |
| PPL | 0.69 | 0.78 | 0.67 | 0.67 | 0.65 | 0.68 | 0.69 | 0.64 | 0.71 | 0.59 | 0.64 | 0.62 | 0.68 | 0.65 |
| Passage Utility | 0.74 | 0.80 | 0.71 | 0.78 | 0.91 | 0.75 | 0.78 | 0.75 | 0.77 | 0.70 | 0.78 | 0.88 | 0.78 | 0.78 |

Table 17: AUROC values for the Passage Utility and perplexity baseline on individual passages on the six test sets (NQ, TQA, WebQ, SQuAD, PopQA, and RefuNQ).

| | NQ | | TQA | | WebQ | | SQuAD | | PopQA | | RefuNQ | |
|---|---|---|---|---|---|---|---|---|---|---|---|---|
| Development | Acc | AccLM | Acc | AccLM | Acc | AccLM | Acc | AccLM | Acc | AccLM | Acc | AccLM |
| Gemma2-9B | 0.48 | 0.66 | 0.74 | 0.80 | 0.46 | 0.66 | 0.38 | 0.60 | 0.51 | 0.52 | — | — |
| Gemma2-27B | 0.48 | 0.66 | 0.75 | 0.81 | 0.49 | 0.67 | 0.38 | 0.60 | 0.52 | 0.52 | — | — |
| Llama-3.1-8B | 0.48 | 0.62 | 0.71 | 0.77 | 0.53 | 0.64 | 0.39 | 0.57 | 0.51 | 0.49 | — | — |
| Mistral-7B-v0.3 | 0.48 | 0.62 | 0.72 | 0.76 | 0.52 | 0.69 | 0.37 | 0.58 | 0.53 | 0.51 | — | — |

| | NQ | | TQA | | WebQ | | SQuAD | | PopQA | | RefuNQ | |
|---|---|---|---|---|---|---|---|---|---|---|---|---|
| Test | Acc | AccLM | Acc | AccLM | Acc | AccLM | Acc | AccLM | Acc | AccLM | Acc | AccLM |
| Gemma2-9B | 0.49 | 0.65 | 0.74 | 0.80 | 0.40 | 0.66 | 0.43 | 0.60 | 0.50 | 0.52 | 0.26 | 0.40 |
| Gemma2-27B | 0.48 | 0.65 | 0.76 | 0.81 | 0.41 | 0.66 | 0.42 | 0.61 | 0.51 | 0.53 | 0.26 | 0.39 |
| Llama-3.1-8B | 0.49 | 0.61 | 0.71 | 0.77 | 0.44 | 0.63 | 0.43 | 0.58 | 0.50 | 0.49 | 0.27 | 0.36 |
| Mistral-7B-v0.3 | 0.49 | 0.62 | 0.72 | 0.77 | 0.47 | 0.66 | 0.41 | 0.58 | 0.51 | 0.50 | 0.26 | 0.35 |

Table 18: QA model performance (with $|R| = 5$) on the development and test sets. We report token- and model-based accuracy (Acc and AccLM). AccLM is computed by Qwen2-72B-Instruct.

### G.3 Detailed Uncertainty Estimation Results

Table 19 shows AURAC scores on the test sets. Table 20 shows the performance of uncertainty quantification approaches on the development set. We report AUROC and AURAC.

## H Examples of False Positives and Negatives

Tables 21–24 illustrate the working of Passage Utility for answer uncertainty estimation. As we report AUROC scores, we do not set any correct/incorrect decision threshold; for the purpose of this discussion, we assume a decision point at 0.5 and analyze clear success and failure cases. For each example, we show the question, gold, and generated answers in the top block. Then, we show three retrieved passages with their estimated Passage Utility and a final block with ten sampled answers, their grouping into clusters, and the Cluster Assignment entropy.

Table 21 shows an example for a SQuAD question and the Llama-3.1-8B QA model. In this case, the QA model correctly answers and the Passage Utility estimate is high (i.e., indicating the answer is correct). Table 22 illustrates a case where Llama-3.1-8B's answer is incorrect and all Passage Utilities are very low (i.e., indicating the answer is incorrect). The example from NQ in Table 23 shows a case where all Passage Utilities are low but the QA model (Gemma2-9B) answers correctly. The first passage is not useful, the second does not explicitly mention the answer but still primes the QA model to answer correctly, while the third passage mentions the answer.

| | Gemma2-9B | | | | | | Gemma2-27B | | | | | |
| --- | --- | --- | --- | --- | --- | --- | --- | --- | --- | --- | --- | --- |
| | NQ | TQA | WebQ | SQuAD | PopQA | RefuNQ | NQ | TQA | WebQ | SQuAD | PopQA | RefuNQ |
| PPL | 0.69 | 0.84 | 0.63 | 0.57 | 0.56 | 0.45 | 0.67 | 0.78 | 0.63 | 0.62 | 0.56 | 0.45 |
| p(true) | 0.75 | 0.85 | 0.71 | 0.63 | 0.71 | 0.53 | **0.76** | **0.89** | **0.73** | 0.67 | 0.70 | 0.54 |
| Regular Entropy | 0.70 | 0.84 | 0.63 | 0.59 | 0.57 | 0.46 | 0.69 | 0.79 | 0.64 | 0.61 | 0.58 | 0.45 |
| Semantic Entropy | 0.71 | 0.85 | 0.64 | 0.65 | 0.64 | 0.51 | 0.69 | 0.81 | 0.67 | 0.63 | 0.61 | 0.50 |
| Passage Utility | **0.76** | **0.90** | **0.72** | **0.74** | **0.73** | **0.64** | 0.72 | 0.88 | **0.73** | **0.74** | **0.73** | **0.64** |

| | Llama-3.1-8B | | | | | | Mistral-7B-v0.3 | | | | | |
| --- | --- | --- | --- | --- | --- | --- | --- | --- | --- | --- | --- | --- |
| | NQ | TQA | WebQ | SQuAD | PopQA | RefuNQ | NQ | TQA | WebQ | SQuAD | PopQA | RefuNQ |
| PPL | 0.73 | 0.87 | 0.71 | 0.68 | 0.69 | 0.54 | 0.67 | 0.83 | 0.66 | 0.66 | 0.61 | 0.54 |
| p(true) | **0.76** | **0.89** | **0.75** | 0.70 | **0.71** | 0.59 | 0.71 | 0.86 | 0.70 | 0.69 | 0.65 | 0.56 |
| Regular Entropy | 0.73 | 0.87 | 0.72 | 0.70 | 0.69 | 0.56 | 0.67 | 0.84 | 0.69 | 0.65 | 0.62 | 0.51 |
| Semantic Entropy | 0.71 | 0.87 | 0.71 | 0.70 | 0.67 | 0.54 | 0.68 | 0.85 | **0.71** | 0.69 | 0.66 | 0.51 |
| Passage Utility | 0.74 | 0.87 | 0.73 | **0.73** | **0.71** | **0.65** | **0.72** | **0.87** | **0.71** | **0.75** | **0.71** | **0.66** |

Table 19: AURAC values for QA models based on Gemma2-9B/27B, Llama-3.1-8B, and Mistral-7B-v0.3 on Natural Questions (NQ), TriviaQA (TQA), WebQuestions (WebQ), SQuAD, PopQA, and RefuNQ test sets.

In Table 24, Passage Utility scores are high estimating a correct answer for the TQA test question; however, Gemma2-9B answers with the incorrect magazine name. Note that none of the passages corresponds to the National Geographic magazine but have high token overlap with the question (in particular the first and second passages).

| Gemma2-9B | AUROC | | | | | AURAC | | | | |
|---|---|---|---|---|---|---|---|---|---|---|
| | NQ | TQA | WebQ | SQuAD | PopQA | NQ | TQA | WebQ | SQuAD | PopQA |
| PPL | 0.61 | 0.52 | 0.58 | 0.66 | 0.56 | 0.67 | 0.78 | 0.67 | 0.65 | 0.52 |
| MSP | 0.64 | 0.60 | 0.64 | 0.71 | 0.61 | 0.69 | 0.80 | 0.69 | 0.67 | 0.56 |
| PMI | 0.53 | 0.46 | 0.52 | 0.50 | 0.48 | 0.64 | 0.75 | 0.64 | 0.57 | 0.50 |
| p(true) | 0.70 | 0.71 | 0.66 | 0.73 | 0.83 | 0.72 | 0.84 | 0.70 | 0.69 | **0.71** |
| Regular Entropy | 0.64 | 0.54 | 0.60 | 0.70 | 0.58 | 0.69 | 0.78 | 0.68 | 0.67 | 0.54 |
| Cluster Assignment | 0.68 | 0.65 | 0.65 | 0.70 | 0.68 | 0.71 | 0.82 | 0.70 | 0.67 | 0.60 |
| Semantic Entropy | 0.67 | 0.69 | 0.64 | 0.72 | 0.69 | 0.71 | 0.84 | 0.69 | 0.68 | 0.61 |
| AvgAnsLen | 0.61 | 0.64 | 0.65 | 0.63 | 0.68 | 0.68 | 0.83 | 0.71 | 0.65 | 0.61 |
| Retriever Score | 0.53 | 0.62 | 0.53 | 0.67 | 0.64 | 0.65 | 0.82 | 0.63 | 0.67 | 0.61 |
| Passage Utility | **0.72** | **0.84** | **0.75** | **0.85** | **0.85** | **0.75** | **0.89** | **0.77** | **0.77** | **0.71** |

| Gemma2-27B | AUROC | | | | | AURAC | | | | |
|---|---|---|---|---|---|---|---|---|---|---|
| | NQ | TQA | WebQ | SQuAD | PopQA | NQ | TQA | WebQ | SQuAD | PopQA |
| PPL | 0.61 | 0.56 | 0.55 | 0.63 | 0.53 | 0.68 | 0.79 | 0.65 | 0.67 | 0.52 |
| MSP | 0.64 | 0.66 | 0.59 | 0.67 | 0.60 | 0.70 | 0.82 | 0.67 | 0.69 | 0.56 |
| PMI | 0.51 | 0.52 | 0.56 | 0.54 | 0.56 | 0.64 | 0.78 | 0.67 | 0.62 | 0.56 |
| p(true) | **0.76** | 0.73 | 0.69 | 0.69 | 0.79 | **0.78** | 0.84 | 0.74 | 0.71 | 0.70 |
| Regular Entropy | 0.65 | 0.53 | 0.56 | 0.64 | 0.53 | 0.71 | 0.78 | 0.66 | 0.67 | 0.52 |
| Cluster Assignment | 0.66 | 0.67 | 0.59 | 0.66 | 0.66 | 0.71 | 0.82 | 0.67 | 0.68 | 0.60 |
| Semantic Entropy | 0.64 | 0.67 | 0.59 | 0.68 | 0.66 | 0.69 | 0.82 | 0.68 | 0.69 | 0.60 |
| AvgAnsLen | 0.63 | 0.68 | 0.65 | 0.60 | 0.69 | 0.69 | 0.83 | 0.72 | 0.66 | 0.61 |
| Retriever Score | 0.56 | 0.60 | 0.51 | 0.69 | 0.65 | 0.67 | 0.81 | 0.64 | 0.71 | 0.62 |
| Passage Utility | 0.73 | **0.75** | **0.72** | **0.84** | **0.87** | 0.75 | **0.86** | **0.75** | **0.78** | **0.73** |

| Llama-3.1-8B | AUROC | | | | | AURAC | | | | |
|---|---|---|---|---|---|---|---|---|---|---|
| | NQ | TQA | WebQ | SQuAD | PopQA | NQ | TQA | WebQ | SQuAD | PopQA |
| PPL | 0.75 | 0.78 | 0.68 | 0.75 | 0.81 | 0.76 | 0.85 | 0.71 | 0.71 | 0.68 |
| MSP | 0.77 | 0.80 | 0.71 | 0.76 | 0.85 | 0.76 | 0.85 | 0.72 | 0.72 | 0.70 |
| PMI | 0.55 | 0.52 | 0.48 | 0.54 | 0.58 | 0.64 | 0.73 | 0.60 | 0.61 | 0.53 |
| p(true) | **0.80** | **0.86** | 0.72 | **0.82** | 0.85 | **0.78** | **0.87** | 0.74 | **0.75** | **0.71** |
| Regular Entropy | 0.77 | 0.80 | 0.69 | 0.76 | 0.83 | 0.76 | 0.85 | 0.71 | 0.72 | 0.69 |
| Cluster Assignment | 0.75 | 0.83 | 0.69 | 0.75 | 0.82 | 0.75 | 0.85 | 0.71 | 0.71 | 0.67 |
| Semantic Entropy | 0.74 | 0.83 | 0.70 | 0.74 | 0.81 | 0.75 | 0.86 | 0.72 | 0.71 | 0.68 |
| AvgAnsLen | 0.73 | 0.73 | 0.69 | 0.69 | 0.84 | 0.73 | 0.82 | 0.71 | 0.67 | 0.69 |
| Retriever Score | 0.58 | 0.63 | 0.54 | 0.68 | 0.66 | 0.65 | 0.79 | 0.62 | 0.66 | 0.60 |
| Passage Utility | 0.78 | 0.85 | **0.74** | **0.82** | **0.86** | 0.75 | **0.87** | **0.75** | **0.75** | **0.71** |

| Mistral-7B-v0.3 | AUROC | | | | | AURAC | | | | |
|---|---|---|---|---|---|---|---|---|---|---|
| | NQ | TQA | WebQ | SQuAD | PopQA | NQ | TQA | WebQ | SQuAD | PopQA |
| PPL | 0.66 | 0.70 | 0.60 | 0.63 | 0.66 | 0.69 | 0.84 | 0.72 | 0.63 | 0.63 |
| MSP | 0.70 | 0.75 | 0.65 | 0.71 | 0.77 | 0.70 | 0.85 | 0.73 | 0.68 | 0.67 |
| PMI | 0.38 | 0.33 | 0.42 | 0.42 | 0.30 | 0.53 | 0.68 | 0.62 | 0.52 | 0.39 |
| p(true) | 0.72 | **0.82** | 0.71 | 0.75 | 0.74 | 0.71 | **0.87** | 0.76 | 0.71 | 0.64 |
| Regular Entropy | 0.67 | 0.71 | 0.63 | 0.66 | 0.68 | 0.69 | 0.85 | 0.73 | 0.66 | 0.63 |
| Cluster Assignment | 0.72 | 0.81 | 0.68 | 0.73 | 0.76 | 0.71 | **0.87** | 0.75 | 0.68 | 0.66 |
| Semantic Entropy | 0.72 | 0.80 | 0.68 | 0.73 | 0.76 | 0.71 | **0.87** | 0.76 | 0.69 | 0.66 |
| AvgAnsLen | 0.66 | 0.75 | 0.65 | 0.68 | 0.81 | 0.69 | 0.85 | 0.73 | 0.67 | 0.70 |
| Retriever Score | 0.55 | 0.63 | 0.55 | 0.65 | 0.68 | 0.63 | 0.81 | 0.67 | 0.64 | 0.63 |
| Passage Utility | **0.76** | 0.81 | **0.75** | **0.85** | **0.85** | 0.74 | **0.87** | **0.78** | **0.75** | **0.71** |

Table 20: Answer uncertainty estimation for QA models Gemma2-9/27B, Llama-3.1-8B, and Mistral-7B-v0.3 on NQ, TQA, WebQ, SQuAD, and PopQA development sets. We report AUROC and AURAC.

Question: How many platforms does Manchester Oxford Road have?
Generated Answer: Four through platforms and one terminating bay platform.

Gold Answer: 5.

---

Originally one, later expanded.

Manchester Oxford Road railway station. Gloucester Street (now Whitworth Street West) to reach their north front. There was a single platform on the north side of the line through to Manchester London Road (now "Manchester Pic adilly") and a second platform on a west-facing siding. To allow for extra trains in connection with the Manchester Art Treasures Exhibition in 1857, extra platforms and sidings were bui lt, but afterwards removed. A meeting of MSJAR contract ticket holders in 1863 included in its list of complaints the want of punctuality "especially as at Oxford Road station there is only one platform used for both passengers and milk".  -0.42

---

Twelve terminating, two through.

Transport in Manchester. platforms are planned as part of the Northern Hub project however in 2017 the construction was not confirmed. At present there are twelve terminating platforms but just two through platforms. Congestion is common at peak times and trains often incur delays for following trains down the line at Oxford Road. The new proposals will provide up to 700 more trains per day with space for 44 million more people to travel by train each year. Victoria station has undergone a series of upgrades. The station was rated as the worst in the UK following a 2009 study and plans for  -0.26

---

Five platforms.

Manchester Oxford Road railway station. Manchester Oxford Road railway station is a railway station in Manchester, England, at the junction of Whitworth Street West and Oxford Street. It opened in 1849 and was rebuilt in 1960. It is the second busiest of the four stations in Manchester city centre. The station serves the southern part of Manchester city centre, the University of Manchester and Manchester Metropolitan University, on the line from westwards towards Warrington, Chester, Llandudno, Liverpool, and Blackpool. Eastbound trains go beyond Piccadilly to , and . The station consists of four through platforms and one terminating bay platform. 1.34

---

['Four through platforms and one terminating bay platform.', 'Four through platforms and one terminating bay.', 'Four through platforms and one terminating bay platform.'], ['Four through and one terminating.', 'Four through platforms and one terminating bay.', 'Four through and one bay.'], ['Five platforms.', 'Five platforms.', 'Five platforms.', 'Five platforms.']  1.09

---

Table 21: True negative example (from NQ development set): Passage Utility predicts the right answer as well as the QA model (GEMMA2-9B).

Question: Which company was targeted by the NAACP for not having fair practices?
Generated Answer: Target Corporation.

Gold Answer: Lockheed Aircraft Corporation.

---

```
Target Corporation.
```

```
Target Corporation. of Colored People has repeatedly given Target failing grades on its annual Economic Reciprocity Initiative
report card, a measure of the company's "commitment to the African-American citizenry". In 2003 and 2005, the NAACP has rated Target
an "F" on this report; in 2004, Target was rated a "D-". In 2006, when Target was asked why it didn't participate in the survey
again, a representative explained, "Target views diversity as being inclusive of all people from all different backgrounds, not just
one group." In February 2006, the National Federation of the Blind (NFB) filed a class action [cont.]                     -2.64
```

```
None, the NAACP was involved in the Duke lacrosse case.
```

```
Reactions to the Duke lacrosse case. formed an opinion on the case. North Carolina NAACP Legal Redress Chair, Al McSurely, explained
that "The NAACP stands for fair play for all parties, zealous investigation and deep concern for the survivors of racist/sexist
attacks." At the same time, some have criticized the NAACP for making statements that portrayed the players as racist despite
evidence to the contrary, using the case to promote the group's cause, and implying guilt. McSurely stated that "[w]ithin five
minutes, the men threatened the women with racial and misogynist verbal assaults, [cont.]                               -2.77
```

```
Philadelphia Transit Company (PTC).
```

```
Philadelphia transit strike of 1944. (PRTEU), Frank Carney, proved to be equally reticent and claimed that he was not authorized by
the union members to consider a request to allow promotions of black employees. The black PTC employees enlisted the help of the NAACP
and started lobbying the federal authorities, particularly the Fair Employment Practices Commission (FEPC), to intervene. The Fair
Employment Practices Commission, created by an executive order of the President in 1941, was charged with ensuring non-discrimination
employment practices by government contractors. [cont.]                                                                  -2.60
```

```
['Target.'], ['Target Corporation.', 'Target Corporation.', 'Target Corporation.', 'Target Corporation.', 'Target Corporation.',
'Target Corporation.', 'Target Corporation.', 'Target Corporation.'], ['Target Corporation and the National Federation of the Blind
filed a lawsuit against Michigan.']                                                                                       0.33
```

Table 22: True positive example (from SQuAD development set): Passage Utility predicts the answer provided by the QA model (LLAMA-3.1-8B) which is incorrect.

Question: What do Alfred Newman, John Williams, Johnny Green, Andr Previn, John Barry and Alan Menken have in common?
Generated Answer: They are all Oscar-winning film composers.

Gold Answer: Winning the Academy award for Best Original Score.

They all won multiple Academy Awards for film scores.

Alan Menken. his other film musicals were also nominated: Alan Menken Alan Irwin Menken (born July 22, 1 949) is an American musical theatre and film score composer and pianist. Menken is best known for his scores for films produced by Walt Disney Animation Studios. His scores for "The Little Mermaid" (1989 ), "Beauty and the Beast" (1991), "Aladdin" (1992), and "Pocahontas" (1995) have each won him two Academy Awards. He also composed the scores for "Little Shop of Horrors" (1986), "Newsies" (1992), "The Hunchback of Notre Dame" (1996), "Hercules" (1997), "Home on the Range" (2004), "Enchanted" (2007), "Tangled" (2010), and "Sausage Party" (2016) -2.22

They are all prolific Oscar winners in music categories.

Alan Menken. hi s work on musical theatre works for Broadway and elsewhere. Some of these are based on his Disney films, but other stage hits include "Little Shop of Horrors" (1982), "A Christmas Carol" (1994) and "S ister Act" (2009). Menken has collaborated with such lyricists as Howard Ashman, Tim Rice, Glenn Slater, Stephen Schwartz and David Zippel. With eight Academy Award wins (four each for Best Score and Best Song), Menken is the second most prolific Oscar winner in the music categories after Alfred Newman, who has nine Oscars. He has also won eleven Grammy Awards, a Tony Award and other honors. -2.30

They are all Academy Award-winning composers.

Alan Menken. the Beast", with the so ngs from the 1991 film and new material by Menken and Rice. , Menken is collaborating on writing new songs with Benj Pasek and Justin Paul for a live-action film version of "Aladdin" and is also working with Lin-Manuel Miranda on new music for a live-action film adaptation of "The Little Mermaid". Menken will also once again be working with Stephen Schwartz to write new songs for D̈isenchanted," the s equel to "Enchanted". With eight Academy Awards (four each for best score and best song), only composer Alfred Newman (nine wins) and Walt Disney (22 wins) have -2.31

['They are all Academy Award-winning composers.', 'They are all Academy Award-winning composers.', 'They are all Oscar-winning film composers.', 'They are all Oscar-winning film composers.'], ['They are all multiple-time Oscar winners for music.', 'They are prolific Academy Award-winning composers.'], ['They are all multi-Oscar winning film composers.', 'They are all multiple Oscar-winning composers.', 'They are prolific Oscar-winning film composers.'], ['They are all prolific film music composers.'] 1.28

Table 23: False positive example (from TQA development set): Passage Utility predicts a wrong answer whereas the prediction of the QA model (Gemma2-9B) is correct.

Question: One Step Behind' is a tribute act to which band?
Generated Answer: Furniture.

Gold Answer: North London Invaders.

---

Def L eppard.

---

Two Steps Behind. Two Steps Behind "Two Steps Behind" is a 1993 song by English hard rock band Def Leppard from their album "Retro Active" and "Last Action Hero" Soundtrack. It reached #5 on the "Billboard" Hot Mainstream Rock Tracks chart, and #12 on the BillboardHot 100. Though the band did manage to chart a few more songs in the following years, this is generally considered to be the band's last major hit in the US. In the 1993 "Metal Edge" Readers' Choice Awards, the song was voted "Song of the Year" and "Best Song From a Movie Soundtrack.". Def Leppard have                                                         3.29

---

Def Leppard.

---

Two Steps Behind. and a live footage. The video was aired on August 19 93. Filipino-Chinese singer Rachelle Ann Go covered the song for her 2007 album "Obsession". Two Steps Behind "Two Steps Behind" is a 1993 song by English hard rock band Def Leppard from their album "Retro Active" and "Last Action Hero" Soundtrack. It reached #5 on the "Billboard" Hot Mainstream Rock Tracks chart, and #12 on the "Billboard" Hot 100. Though the band did manage to chart a few more songs in the following years, this is generally considered to be the band's last major hit in the US. In the                                                         2.42

---

Split Enz.

---

One Step Ahead (Split Enz song). unavailable to Australasian markets until 2007 when it became available on iTunes). The video clip to "One Step Ahead" has keyboardist Eddie Rayner performing "Marche sur place", the pantomime illusion walk created by Decroux and Barrault (seen in the 1945 French film Child ren of Paradise) that is the technique Michael Jackson would base his moonwalk on in 1983. One Step Ahead (Split Enz song) "One Step Ahead" is a 1980 song by New Zealand art rock group Split Enz from the ir studio album "Waiata". The song continued the group's success in their move towards their own version of new wave                  -2.93

---

['Furniture', 'Furniture', 'Furniture', 'Furniture', 'Furniture', 'Furniture', 'Furniture', 'Furniture', 'Furniture', 'Furniture']
0

Table 24: False negative (from TQA development set): Passage Utility predicts a correct answer, and the answer by the QA model (Gemma2-9B) is wrong.

