# OpenReview forum: "Uncertainty Quantification in Retrieval Augmented Question Answering"
_TMLR — Accepted by TMLR_

### Review · Reviewer_FhU7 · 2025-07-05

**Summary Of Contributions:**

The paper introduces a passage-level utility prediction approach for uncertainty quantification in retrieval-augmented QA systems. This method provides a novel uncertainty estimation that can identify which specific passages contribute to unreliable QA performance and offers an alternative to existing answer-level uncertainty methods.

**Audience:**

Yes

**Claims And Evidence:**

Yes

**Requested Changes:**

- While the experimental settings and implementation cite `Farquhar et al. (2024)`, the related work section does not thoroughly discuss how this research relates to the authors' study. I suggest the authors describe the relationship and differences with this research in more detail.

- Can you explain why a simple average of accuracy and entailment is used in Eq.(2)?

- Analysis seems needed on how it works when $υ_{\mathcal{M}}  = a(y)$ and when $υ_{\mathcal{M}}  = e(y)$ in  Eq.(2). Or it would be good if you could report changes according to α values in $υ_{\mathcal{M}} = α·a(y) + (1-α)·e(y)$.

The OOD performance drops in Table 6, and it would be good to analyze this reason further, even if you mentioned leaving it as future work.

- The authors state about HotPotQA results that "This evaluation shows that the idealized multi-hop behavior studied with synthetic passages occurs to a lesser extent in current LLMs." However, doesn't this stem from limitations in the authors' experimental design rather than dataset issues? Particularly, isn't the authors' aggregation method (Eq. 6) inappropriate for multi-hop scenarios?

- As I understand it, since retraining is required whenever the model or task changes, frequent fine-tuning makes it difficult to apply to new QA environments without retraining. Can you discuss more about these limitations?

**Strengths And Weaknesses:**

## Strenghts
- The paper addresses the important problem of uncertainty quantification in retrieval-augmented QA systems.
- The proposed method shows performance improvements on several QA models and datasets.
- The proposed contrastive learning framework with a pairwise ranking loss is a well-designed choice for passage utility prediction. By directly optimizing for relative utility differences between passages conditioned on the same question, the objective function of Eq.(3) enables consistent learning of passage ranking.

## Weaknesses

- The proposed Utility Ranker is a passage utility predictor trained according to the output behavior of the QA model M. When switching to a new QA model, the existing utility predictor becomes non-reusable because it requires retraining for each model change, which creates a weakness of frequent fine-tuning requirements.

- It may appear to combine existing research areas, such as passage quality assessment and uncertainty quantification.

- The explanation for the passage utility definition in Equation 2 is unclear.

- The analysis and explanation of the HotPotQA experiment are sparse compared to other experiments.

- Unlike entropy-based or p(true)-style estimators directly based on model outputs or self-judgment, the utility ranker relies on supervised learning based on QA results and entailment judgments, making it seem dependent on models and tasks.

- Although the contrastive ranking objective is well designed to model relative phrase utility, the pairwise comparison approach appears unlikely to guarantee global ranking consistency.

---

> ### Author Response · Authors · 2025-07-22
> **RE: Weaknesses**
>
> Thank you for the time to read the paper and providing very thoughtful feedback.
>
> **W1**: The proposed Utility Ranker is a passage utility predictor trained according to the output behaviour of the QA model M. When switching to a new QA model, the existing utility predictor becomes non-reusable because it requires retraining for each model change, which creates a weakness of frequent fine-tuning requirements.
>
> Discussed in question Q6 below.
>
>
> **W2**: It may appear to combine existing research areas, such as passage quality assessment and uncertainty quantification.
>
> As discussed in the Related work section (second paragraph about passage quality), previous research in the context of Retrieval Augmented Generation (RAG) often relies on external judges, such as Natural Language Inference (NLI) models, to assess passage quality. While this is helpful for creating training data, as seen in methods like Self-RAG, it's not always reliable for gauging how uncertain a Question Answering (QA) model is.
>
> Our method, which uses a separate model to estimate uncertainty, builds upon techniques previously applied to classification tasks like image recognition or reading comprehension. We've specifically adapted it for retrieval-augmented QA. This is a more challenging domain because the input can be noisy and the number of input passages can vary.
>
> More broadly, our work addresses the challenge of uncertainty quantification in retrieval-augmented QA. Existing methods in this area are often expensive to implement and can suffer from overconfidence or sensitivity to the input context, as highlighted in [5]. In fact, the analysis in [5] regarding how retrieved evidence influences existing uncertainty estimation methods aligns with our own experimental findings: sampling-based methods show less variation when used in a RAG context.
>
> [5] Why Uncertainty Estimation Methods Fall Short in RAG: An Axiomatic Analysis (Soudani et al., 2025)
> https://arxiv.org/abs/2505.07459
>
>
> **W3**: The explanation for the passage utility definition in Equation 2 is unclear.
>
> Discussed in question Q2 below.
>
> **W4**: The analysis and explanation of the HotPotQA experiment are sparse compared to other experiments.
>
> Discussed in question Q5 below.
>
>
> **W5**: Unlike entropy-based or p(true)-style estimators directly based on model outputs or self-judgment, the utility ranker relies on supervised learning based on QA results and entailment judgments, making it seem dependent on models and tasks.
>
> Each uncertainty quantification approach comes with its own advantages and limitations, making its selection contingent on specific criteria such as available resources, required latency, and the desired level of control and trustworthiness for the QA system.
>
> For example, in high-stakes applications, a method prioritizing higher rates of false positives (thereby allowing human intervention) and minimizing the risk of overconfident false negatives would be preferable, even if it necessitates dedicated training data and ongoing updates.
>
> Furthermore, our approach demonstrates competitive performance even when its utility predictor is trained solely on the target model's accuracy, as detailed in the ablation study on the training objective (see appendix). By its very design, our method is inherently dependent on the target QA model, as its core aim is to model that model's accuracy behaviour.
>
> It is important to note that the efficacy of p(true) is significantly influenced by several task-relevant few-shot examples. Consequently, its performance degrades when evaluated in out-of-distribution scenarios, a finding now presented in Table 7 of the updated PDF.
>
> To clarify the trade-offs, challenges, and possible mitigations for different methods, we've added a new Section 6 to the revised PDF.
>
> **W6**: Although the contrastive ranking objective is well designed to model relative phrase utility, the pairwise comparison approach appears unlikely to guarantee global ranking consistency.
>
> While we acknowledge that training with an objective for global ranking consistency could enhance results, it would escalate the training cost. Instead, our use of a binary cross-entropy objective partially fosters global consistency by encouraging higher scores for useful passages and lower scores for useless ones. Although the resulting predictive rankings aren't flawless, they remain valuable for uncertainty estimation.

---

> ### Author Response · Authors · 2025-07-22
> **RE: Requested Changes (Part 1)**
>
> **Q1**: While the experimental settings and implementation cite Farquhar et al. (2024), the related work section does not thoroughly discuss how this research relates to the authors' study. I suggest the authors describe the relationship and differences with this research in more detail.
>
> Semantic entropy (Kuhn et al., 2023; Farquhar et al., 2024) focuses on detecting incorrect answers stemming from arbitrary fluctuations in model outputs (referred to as confabulations).
> Our approach extends to additional error sources, including incorrect training data or misleading evidence. Furthermore, aligning with findings in [5] for retrieval-augmented QA, our empirical evidence suggests that sequence entropy methods do not offer a significant advantage over perplexity or p(true) in this context, unlike in closed-book settings.
>
> In addition to the discussion of differences in terms of latency at inference time, we will also make clearer the differences in terms of implementation requirements, i.e., while semantic entropy requires sampling from the target QA model, Passage Utility requires generating training data with observations of the accuracy behaviour of the target model. Both use NLI models while semantic entropy always needs an NLI model on short answers at inference time, Passage Utility uses (optionally depending on the chosen objective) an NLI model at training time.
>
> Please let us know if there are any further discussion points you feel we overlooked.
>
> **Q2**: Can you explain why a simple average of accuracy and entailment is used in Eq.(2)?
>
> Our aim is to create a single utility score that effectively captures diverse scenarios. This includes instances where the model answers correctly, regardless of the passage's contribution (high or low), as well as situations where it answers incorrectly, even if the passage is highly relevant or completely irrelevant.
>
> To achieve a score between 0 and 1, we initially calculated a simple average. However, for our pairwise ranking loss, using the sum would be equivalent. We'll clarify this equivalence in the description of Equation (2) in the revised PDF.
>
> **Q3**: Analysis seems needed on how it works when and when in Eq.(2). Or it would be good if you could report changes according to α values in variants of [$\alpha$ * a + (1- $\alpha$) * e] in the ranking loss.
>
> In Table 4 (appendix of the new version of the pdf), we show an ablation study on the contribution of the two component losses of the training objective. When looking at rows 1-3, we observe that the contribution of the binary cross-entropy is strong w/ the pairwise ranking bringing improvements for some models. Rows 1-2 (ranking based on entailment alone vs entailment and accuracy) show that when trying to optimise ranking with only entailment and the binary cross-entropy objective performance drops. In other words, our results show that entailment judgements by an external judge are not an optimal predictor of the QA model accuracy behaviour.
>
> As suggested above, we carry out complementary ablation experiments on different variants of the pairwise ranking loss on the Gemma2-9B model and the WebQ dataset (below the results). We will add these to the revised PDF.
>
>
> | Objective        | AUROC |
> |------------------|-------|
> |(e+a)/2 + BCE     | 0.75  |
> |(e+a)   + BCE     | 0.74  |
> |(e∨(1-e)+a) + BCE | 0.72  |
> |(e) + BCE         | 0.63  |
> |BCE               | 0.71  |
> |(e+a)/2           | 0.65  |
> |(e+a)             | 0.63  |
> |(e∨(1-e)+a)       | 0.62  |
> |(e)               | 0.61  |
> |(a)               | 0.53  |

---

> ### Author Response · Authors · 2025-07-22
> **RE: Requested Changes (Part 2)**
>
> **Q4**: The OOD performance drops in Table 6, and it would be good to analyse this reason further, even if you mentioned leaving it as future work.
>
> (now Table 7 in the updated version of the PDF) To understand the observed performance drop, we conducted a comparative analysis of passage utility predictions for 50 question-passage pairs using the Gemma2-9B QA model. We examined predictions from a utility predictor trained on WebQ data (in-distribution) versus one trained on SQuAD (out-of-distribution).
>
> We sampled 10 WebQ test questions: 5 where the predicted utility shifted from high in-distribution to low out-of-distribution (a change from true negative to false positive), and 5 where it shifted from low in-distribution to high out-of-distribution (a change from true positive to false negative). In both scenarios, the out-of-distribution predictions were predominantly influenced by token overlap and similarity. We hypothesize that when faced with out-of-distribution questions (e.g., in terms of type, length, or topic), the prediction mechanism becomes more reliant on the degree of token overlap rather than an accurate understanding of the QA model's accuracy behavior. For example, passages with low entailment that nonetheless yield correct answers receive low utility scores in the out-of-distribution setting.
>
> We will include this analysis in Section D of the Appendix.
>
> **Q5**: The authors state about HotPotQA results that "This evaluation shows that the idealized multi-hop behaviour studied with synthetic passages occurs to a lesser extent in current LLMs." However, doesn't this stem from limitations in the authors' experimental design rather than dataset issues? Particularly, isn't the authors' aggregation method (Eq. 6) inappropriate for multi-hop scenarios?
>
> Our approach estimates uncertainty by selecting the maximum passage utility score from sets of retrieved passages. A potential limitation of this method arises with certain multi-hop questions where the necessary evidence is distributed across multiple passages. Specifically, the model might struggle to answer correctly when prompted with individual "hop" passages (resulting in low passage utility), yet succeed when presented with the complete set of passages.
>
> To quantify this limitation, we conducted experiments using the widely-adopted HotPotQA dataset, maintaining the same retrieval settings as our main experiments. The results indicate that our approach performs comparably to existing methods and outperforms the perplexity baseline.
>
> Further manual inspection of passage utility and correctness annotations in the development set revealed two key trends regarding multi-hop behaviour:
>
>  1. Large Language Models (LLMs) frequently manage to answer multi-hop questions using only one of the required hop-passages, often not needing the entire set. This observation aligns with recent findings in [1].
>
>  2. In numerous instances, all passages within a set exhibited low utility, leading the model to answer incorrectly even when prompted with the full set. This underscores the inherent difficulty of effective retrieval for complex questions.
>
> Many studies, such as [2, 3, 4], tackle this challenge through sophisticated QA pipelines. Notably, some approaches decompose complex questions into sub-questions that can be answered independently from individual passages. In such multi-hop pipelines, our approach could be effectively applied at the sub-question level.
>
>
> [1] Sufficient Context: A New Lens on Retrieval Augmented Generation Systems (Joren et al., 2025)
> https://openreview.net/forum?id=Jjr2Odj8DJ
>
> [2] Adaptive-RAG: Learning to Adapt Retrieval-Augmented Large Language Models through Question Complexity (Jeong et al., 2024)
> https://aclanthology.org/2024.naacl-long.389.pdf
>
> [3] Interleaving Retrieval with Chain-of-Thought Reasoning for Knowledge-Intensive Multi-Step Questions (Trivedi et al., 2023)
> https://aclanthology.org/2023.acl-long.557.pdf
>
> [4] Explore What LLM Does Not Know in Complex Question Answering (Lin et al., 2025)
> https://ojs.aaai.org/index.php/AAAI/article/view/34638
>
>
> **Q6**: As I understand it, since retraining is required whenever the model or task changes, frequent fine-tuning makes it difficult to apply to new QA environments without retraining. Can you discuss more about these limitations?
>
> While the utility predictor necessitates recalibration for distinct target QA models or tasks, its performance in out-of-distribution scenarios (Appendix D) establishes it as a robust warm-up base model. This enables subsequent fine-tuning with a reduced number of target task examples.

---

> > ### Comment · Reviewer_FhU7 · 2025-08-06
> >
> > Thank you for addressing my questions.
> >
> > I have reviewed the ablation study for response to Q2. Regarding Q1, I appreciate your explanation of the differences between semantic entropy. It would be helpful for my rating if you could provide specific plans or details on how you will make these differences clearer in the paper. For example, using a table to clarify the differences might be a good idea. For response to Q5, the background knowledge and relevant references you provided about individual passages helped me better understand the context.
> >
> > Regarding Q4, I'm curious whether "passages with low entailment that nonetheless yield correct answers..." represents a limitation of your method or an issue of insufficient training data. Additionally, could you further explain which parts empirically confirm your hypothesis about token overlap?
> >
> > For Q5, it would be beneficial if the paper could further address the limitations of your max aggregation (Eq. 6). Alternatively, I suggest analyzing HotPotQA by dividing it into groups: 1)solvable with a single passage, 2)requiring multiple passages, and 3) insufficient passages.

---

> ### Author Response · Authors · 2025-08-11
>
> Thank you for your response.
>
> **Q1**: Clear discussion of differences with Semantic Entropy.
>
> We propose to extend the discussion in the Related Work section (see updated PDF purple-colored text in Section 2) and further highlight differences in Table 3 and second paragraph in Section 6.
>
> **Q4**: "passages with low entailment that nonetheless yield correct answers..." limitation or insufficient training data. Additionally, could you further explain which parts empirically confirm your hypothesis about token overlap?
>
> We hypothesize that this is due to insufficient training. Analysis of incorrect o.o.d cases revealed that the Passage Utility predictor defaults to predictions based on a general notion of question-passage similarity (i.e., if the passage has low similarity/relevance to the question, then it will predict low utility), disregarding whether the QA model can answer the question with the given passages or not. This behaviour may be a consequence of under-training, suggesting that the Passage Utility predictor has not learned the answering accuracy behaviour of the target QA model on o.o.d questions and their corresponding retrieved passages.
>
> To empirically validate these observations, we calculate Spearman's rho correlation coefficients between the predicted Passage Utility scores and the corresponding retriever relevance scores (as computed by Contriever) for each passage, examining correlations separately for both in-distribution and out-of-distribution predictions. We use the Retriever Score because it measures the semantic relation between questions and retrieved passages. In agreement with our manual inspection findings, there is a positive correlation between  Passage Utility predictions and  retriever scores in the o.o.d setting, suggesting that Passage Utility falls back to textual similarity. In contrast,  the i.i.d setting exhibits a negative and weaker correlation, indicating that training successfully aligns Passage Utility predictions with the target QA model's actual performance rather than superficial retriever similarity scores.
>
> | Semantic Passage Relevance | i.i.d | o.o.d |
> |------------------------------------------|------|--------|
> | Retriever Score | -0.318$^*$ | 0.536$^+$ |
>
>  $^*$ p-value < 0.05,
>  $^+$ p-value < 0.01
>
> Clarification: we use the term 'token overlap' in our previous response because, unfortunately, similarity/relevance/NLI measures are often overreliant on token-level features (e.g., common and proper names, dates, etc.) [1].
>
> [1] Sources of Hallucination by Large Language Models on Inference Tasks
> https://aclanthology.org/2023.findings-emnlp.182/
>
>
> **Q5**: It would be beneficial if the paper could further address the limitations of your max aggregation (Eq. 6). Alternatively, I suggest analyzing HotPotQA by dividing it into groups: 1)solvable with a single passage, 2)requiring multiple passages, and 3) insufficient passages.
>
> We propose the following qualitative analysis given the small number of questions requiring multiple passages in HotPotQA.
>
>
> To understand the behavior of Passage Utility on questions that are only answerable with multiple passages, we analyze multi-hop cases in HotPotQA with Gemma2-9B. Specifically, we look at latent multi-hop and synthetic multi-hop questions. Latent multi-hop questions are selected based on the accuracy of the QA model on individual retrieved passages versus on the full set thereof. That is, cases where the QA model cannot answer when prompted with any individual passage, but answers correctly when prompted with the full set. This suggests that the QA model may be performing some form of reasoning over the set of passages (but not necessarily correspond with the original, manually chosen hop passages released with HotPotQA).  We identified 11 questions with latent multi-hop in the test set. In 6/11 questions, our approach fails to estimate that the QA model will correctly answer with the set of passages.
>
> For synthetic questions, we use the sets of gold passages provided in the HotPotQA dataset to categorise question types as follows. We prompt the model with individual gold passages and with combinations thereof, measuring accuracy. Out of the 500 test questions, 475 are answerable with only one gold 'hop' passage, 13 require multiple passages, and 12 are unanswerable. We then take as synthetic multi-hop questions those where the QA model can only answer based on a combination of passages. In this setting, we insert gold 'hop' passages within the retrieved set for those questions where retrieval failed. Then we run QA and uncertainty estimation. In 6/13 cases, our approach fails to estimate that the answer by the QA model will be correct.
>
> These results suggest that although our approach predicts some cases when the QA model will correctly answer with the 'hop' passages, there is still room for improvement.

---

> > ### Comment · Reviewer_FhU7 · 2025-08-12
> >
> > Thank you for addressing my additional questions.
> >
> > I noticed in Table 11 that Llama-3.1-8B has 790,862 instances for NQ, which is roughly twice as many as the other models. Is this a typo, or was a different configuration used for this model-dataset combination?
> >
> > Regarding the synthetic experiments with gold passages artificially inserted, the authors observe 6 failures out of 11 questions. I'm curious about how this success rate (5/11) is perceived within the research community for this type of analysis, especially given the artificial advantage of having gold passages available.

---

> > > ### Author Response · Authors · 2025-08-12
> > >
> > > **A)** I noticed in Table 11 that Llama-3.1-8B has 790,862 instances for NQ, which is roughly twice as many as the other models. Is this a typo, or was a different configuration used for this model-dataset combination?
> > >
> > > The utility predictor for this specific combination of QA model (Llama-3.1-8b)  and  task (NQ) was trained on the entire training set (as indicated in the last paragraph of Section E.2.
> > >
> > > **B)** Regarding the synthetic experiments with gold passages artificially inserted, the authors observe 6 failures out of 11 questions. I'm curious about how this success rate (5/11) is perceived within the research community for this type of analysis, especially given the artificial advantage of having gold passages available.
> > >
> > > This success rate confirms the limitation of our approach discussed in Q5 (and added into the updated PDF), i.e., it fails at cases where the QA model cannot answer when prompted with any individual passage, but answers correctly when prompted with the full set (false positives). Moreover, this rate is only on the 2.6% of the total cases (13/500).
> > >
> > > We would like to note that gold passages are inserted to create multi-hop scenarios where our approach can fail, in order to expose its  limitations.  When relying on actual retrieved passages, there are no hops in the input set  and the QA model answers incorrectly.
> > >
> > > As discussed in Section 5 "Passage Utility in multi-hop QA", there are two scenarios in retrieval augmented multi-hop QA (given existing datasets). Either models can answer with  one passage (thus no multi-hop) or retrieval completely fails. This is why we resorted to synthetic data  to be able to assess limitations of our approach in the multi-hop QA setting. It is important to note, that in our synthetic  experiment with HotPotQA and Gemma2-9B, the QA model mostly  answers correctly (in 475/500 instances) when it is  prompted  with individual gold  passages.

---

> > > > ### Comment · Reviewer_FhU7 · 2025-08-13
> > > >
> > > > Thank you for the clarifications.
> > > >
> > > > Your explanations have addressed my concerns. I now better understand the design choices and limitations of your approach. If you plan to add any additional sections, tables, or analyses to the revised manuscript based on our discussion (in addition to what you've already mentioned, e.g., Table 3, Section 6, and Appendix D), please let me know.
> > > >
> > > > Thank you again for your responses.

---

### Review · Reviewer_f1fE · 2025-07-14

**Summary Of Contributions:**

This paper introduces a new approach for assessing uncertainty in retrieval-augmented question answering. The authors hypothesize that the usefulness of each retrieved passage reflects the model’s uncertainty in producing the correct answer. To leverage this, they propose training a compact model to estimate passage utility, which is then used to predict answer uncertainty. The method is lightweight, efficient, and effective, and is evaluated across multiple QA datasets with comprehensive baseline comparisons.

**Audience:**

Yes

**Claims And Evidence:**

Yes

**Requested Changes:**

Some of the requested changes are directly related to the previously mentioned weaknesses. In addition, some aspects of the presentation could be improved to enhance clarity.
  - In Equation 4, there are two instances of $ p $ in $ p(x, p) $, which is somewhat unusual. It might be clearer to use a different notation for the inner variable, such as $ p_1 $ or $ p_a $.
  - In Table 1, the term "Utility Ranker" is used in the experimental results without being formally defined in the text.
  - In Figure 2, it is unclear which data points correspond to which methods, as the legend lacks sufficient detail.

**Strengths And Weaknesses:**

Strengths
- The paper is clearly written and easy to understand.
- The proposed method is innovative, and sufficient technical details are introduced.
- The approach is thoroughly evaluated with extensive performance analysis.

Weaknesses
- My primary concern relates to the dataset used for training the passage utility predictor. The paper currently does not provide sufficient details about this dataset. Including statistics and a more comprehensive description of the curated dataset would be valuable and enhance the paper’s clarity.
- A separate passage utility predictor needs to be trained for each base model and task, which is a limitation of the current method and experimental setup. It raises the question of whether it is possible to develop a more general passage utility predictor that can be shared across different models and tasks. Minor comment: In this context, the passage utility predictor is analogous to a reward model, which is often trained using the Bradley-Terry (BT) loss. It would also be interesting to explore alternative loss functions for training of the passage utility predictor.
- Since the overall performance of the method depends on the effectiveness of the passage utility predictor, it is important to include a separate evaluation of the predictor’s performance. This would provide valuable insight into how well the predictor functions and its impact on the final results.

---

> ### Author Response · Authors · 2025-07-23
> **RE: Weaknesses and Requested Changes**
>
> Thank you for the time to read the paper and providing very thoughtful feedback.
>
> **W1**: Details about the curated dataset to train the passage utility predictor.
>
> We added details in the appendix, Section.E.2 paragraph entitled 'Curated Passage Utility Dataset'.
>
> **W2**: A separate passage utility limitation. Possibility of a more general passage utility predictor. Passage utility predictor analogous to a reward model. Bradley-Terry (BT) loss and alternative loss functions.
>
> Our approach, by design, requires fine-tuning to adapt to new QA tasks or models, as its core aim is to model the accuracy behaviour of the target QA model. To enhance versatility across QA tasks, a unified training set encompassing diverse QA tasks could be compiled to train a single passage utility predictor. More practically, advanced training schemes promoting generalization, such as meta-learning with a varied set of QA tasks and QA model examples, could be employed to develop a single passage utility predictor.
>
> We agree that it would be interesting to explore additional loss functions such as Bradley-Terry (BT) and variants as well as including contrastive examples.
>
> **W3**: Evaluating the performance of the passage utility predictor alone.
>
> We add results for this evaluation in the appendix Section.F.2 entitled 'Results on Individual Passage Utility Prediction'.
>
> **Q1**, **Q2**, and **Q3**.
>
> Thank you for pointing these, we will fix them in the final PDF version.

---

> > ### Comment · Reviewer_f1fE · 2025-07-26
> > **Concerns well addressed**
> >
> > Thank you for the further clarification and the additional experimental results. My concerns have been well addressed. The details regarding the curated datasets used for training the passage utility predictor would be helpful for the broader community to gain more in-depth insights into the proposed method. Additionally, the new experiments on the predictor demonstrate its effectiveness compared to a naive PPL baseline, which further strengthens the overall understanding regarding the performance improvement of the framework.

---

### Review · Reviewer_r8hv · 2025-07-15

**Summary Of Contributions:**

The authors introduce an intuitive approach for quantifying the uncertainty of LLM-generated responses within a retrieval-augmented QA framework. Rather than relying on multiple sample generations, their method predicts the utility of individual retrieved passages via a lightweight neural network and employs this estimate to derive an uncertainty measure for the LLM’s response. Empirical evaluations across six QA benchmarks and three smaller-scale LLMs demonstrate that this strategy outperforms conventional information‑theoretic uncertainty metrics.

**Audience:**

Yes

**Claims And Evidence:**

Yes

**Requested Changes:**

The following are not critical but would strengthen the work:
- Reorganize Section 4 to establish a more coherent narrative flow.
- In Equation 2, it appears that a passage yielding an accurate yet non‑entailing response is assigned identical utility to one that is inaccurate but entails the target. Please clarify if this interpretation is correct, and if so whether this equivalence is intentional. If intentional, provide an intuitive rationale for its desirability.

**Strengths And Weaknesses:**

Strengths:
- The proposed framework is firmly rooted in problem structure: by emphasizing passage utility instead of response variability, the method capitalizes on the inherent retrieval‑QA pipeline. Moreover, avoiding repeated LLM sampling yields substantial advantages in settings constrained by latency or cost.
- For smaller LLMs, the authors provide compelling empirical evidence. The results are clearly presented, and the accompanying discussion effectively reinforces the core claims.
- The exploration of reranking via Passage Utility (Figure 4) is particularly noteworthy. These experiments reveal that utility‑based reranking not only serves as an uncertainty proxy but also enhances QA accuracy relative to standard retriever relevance scores, thereby underscoring the dual applicability of the approach.

Weaknesses:
- The necessity of training a neural predictor introduces a dependency on sufficient data, potentially constraining applicability. As illustrated in Appendix D, out‑of‑distribution generalization remains limited; thus, in low‑resource contexts, the method’s viability may be compromised.
- Section 4’s organization could benefit from revision. Although comprehensive, the presentation occasionally shifts abruptly between topics, which may impede reader comprehension.

---

> ### Author Response · Authors · 2025-07-22
> **RE: Weaknesses and Requested Changes**
>
> Thank you for the time to read the paper and providing very thoughtful feedback.
>
> **W1**: The necessity of training a neural predictor introduces a dependency on sufficient data, potentially constraining applicability. As illustrated in Appendix D, out‑of‑distribution generalization remains limited; thus, in low‑resource contexts, the method’s viability may be compromised.
>
> While our approach requires training data and fine-tuning, it offers benefits that supervision-free methods lack. To clarify the trade-offs, challenges, and possible mitigations for different methods, we've added a new Section 6 to the revised PDF.
>
> **W2**/**Q1**: Reorganisation of Section 4.
>
> We will address this the final version of PDF.
>
> **Q2**: In Equation 2, it appears that a passage yielding an accurate yet non‑entailing response is assigned identical utility to one that is inaccurate but entails the target. Please clarify if this interpretation is correct, and if so whether this equivalence is intentional. If intentional, provide an intuitive rationale for its desirability.
>
> Your interpretation is accurate. In practice, however, such exact equalities are less frequent due to the rarity of extreme entailment values (e.g., 1). Furthermore, during the creation of our curated training dataset for the passage utility predictor, we intentionally discard pairwise instances exhibiting identical utility scores.
>
> Beyond our current design, an improved implementation of the utility score would calculate (e+a) for accurate responses and (e∨(1-e)+a) for inaccurate ones. While the current pairwise ranking already demonstrates strong performance, this alternative design could potentially yield further improvements. We propose to investigate this alternative for Equation (2) in an additional ablation study, as suggested in response to Reviewer FhU7.
>
> Ultimately, a passage leading to an accurate but non-entailing response suggests the model is relying more on its internalized knowledge. Conversely, a passage resulting in an inaccurate yet entailing response points to a highly misleading context, which arguably warrants a stronger penalty than a passage leading to an inaccurate but irrelevant response.

---

> > ### Comment · Reviewer_r8hv · 2025-08-04
> > **Response to Author Response**
> >
> > Thank you for the response and the revision. My concerns are adequately addressed by the addition of section 6 and the promised revision of section 4 in the final version. I appreciate the discussion to Q2 above. Since the presence of extreme entailment is rare, this is not a concern, but rather is just an edge case.

---

### Decision · Action_Editor_K2wg · 2025-08-14

**Recommendation:** Accept as is

**Additional Comments:**

This is a solid paper that achieves its stated goals. The reviewers unanimously recommended acceptance. The main issues are primarily ones of scope: (1) the trained models may not generalize beyond the specific base models they are tailored for; (2) the overall methodological innovation is low. However, it substantiates its claims and is above bar for TMLR!

The authors should revise the draft to prepare a final version given the promised changes here, and highlighted comments can be turned to  black for the final version of the paper!

**Audience:**

Yes

**Audience Explanation:**

Understanding the utility of passages is useful for RAG and other applications of retrievers and language models that are of high contemporary interest

**Claims And Evidence:**

Yes

**Claims Explanation:**

This paper claims to train a model to predict whether a passage will be useful for answering a question based on quantifying the uncertainty when answering given that passage. The paper delivers on this claim and establishes that the passage utility method outperforms several baselines.

---

> ### Author Response · Authors · 2025-09-11
>
> Dear Action Editor and Reviewers,
>
> We have submitted the camera-ready version of our paper. This includes additional experiments, analyses, and discussions which arose from the reviewing process. We believe our paper has been greatly improved, we would like to thank the reviewers and action editor for their time and valuable feedback!
>
> Best regards,
> Authors